# UniCBE: An Uniformity-driven Comparing Based Evaluation Framework with Unified Multi-Objective Optimization

**Peiwen Yuan**[1], **Shaoxiong Feng**[2], **Yiwei Li**[1], **Xinglin Wang**[1], **Yueqi Zhang**[1],
**Jiayi Shi**[1], **Chuyi Tan**[1], **Boyuan Pan**[2], **Yao Hu**[2], **Kan Li**[1‡]

[1] School of Computer Science, Beijing Institute of Technology

[2] Xiaohongshu Inc

{peiwenyuan,liyiwei,wangxinglin,zhangyq}@bit.edu.cn

{shijiayi, tanchuyi, likan}@bit.edu.cn

{shaoxiongfeng2023}@gmail.com  {panboyuan,xiahou}@xiaohongshu.com

## Abstract

Human preference plays a significant role in measuring large language models and guiding them to align with human values. Unfortunately, current comparing-based evaluation (CBE) methods typically focus on a single optimization objective, failing to effectively utilize scarce yet valuable preference signals. To address this, we delve into key factors that can enhance the accuracy, convergence, and scalability of CBE: suppressing sampling bias, balancing descending process of uncertainty, and mitigating updating uncertainty. Following the derived guidelines, we propose UniCBE, a unified uniformity-driven CBE framework which simultaneously optimize these core objectives by constructing and integrating three decoupled sampling probability matrices, each designed to ensure uniformity in specific aspects. We further ablate the optimal tuple sampling and preference aggregation strategies to achieve efficient CBE. On the AlpacaEval benchmark, UniCBE saves over 17% of evaluation budgets while achieving a Pearson correlation with ground truth exceeding 0.995, demonstrating excellent accuracy and convergence. In scenarios where new models are continuously introduced, UniCBE can even save over 50% of evaluation costs, highlighting its improved scalability.

## 1 Introduction

The ongoing evolution of large language models (LLMs) has made it increasingly important to assess their alignment with human preferences (Dubois et al., 2024; Zheng et al., 2023). The preference signals provided by humans are crucial for accurately assessing and guiding models toward safe and reliable AGI (Ji et al., 2023; Jiang et al., 2024). However, the rapid iteration of LLMs in training and application scenarios has created a substantial demand for evaluation, complicating the acquisition of sufficient labor-intensive human preferences (Chiang et al., 2024; Cui et al., 2023). Therefore, exploring the use of precious preference signals for efficient model alignment evaluation is of great significance and requires long-term research.

Current mainstream model evaluation paradigms include scoring-based evaluation (SBE) (Liu et al., 2023; Cai et al., 2023) and comparing-based evaluation (CBE) (Chiang et al., 2024; Dubois et al., 2024). The former requires the judge to offer preference scores for individual responses, while the latter needs the judge to establish a preference order among multiple candidate model responses. By directly comparing the responses of different models, Zheng et al. (2023); Liu et al. (2024) confirm that CBE can more accurately assess model performance. However, the $O(NM^2)$ evaluation overhead limits the practicality of CBE when there are $M$ models to evaluate on $N$ samples (Qin et al., 2024). To achieve efficient CBE, various methods have been explored Chiang et al. (2024); Zhou et al. (2024); Dubois et al. (2024). As shown in Figure 1, based on existing observational results, these methods iteratively allocate preference budget to the next (models, sample) tuple according to

---

[‡]Corresponding author.

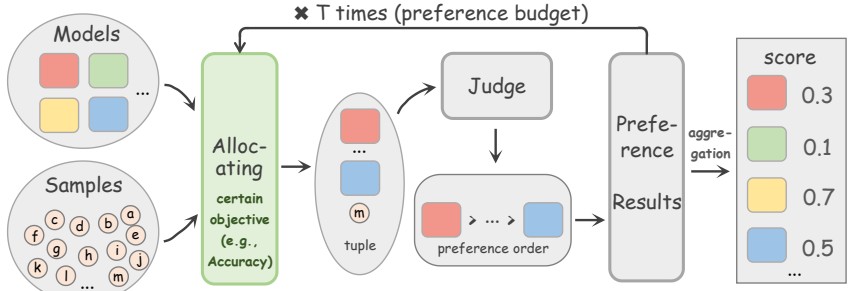

Figure 1: Flowchart of the process for comparing-based evaluation.

respective optimization objectives. Specific preference aggregation methods (e.g., ELO rating (Elo & Sloan, 1978)) are then applied to predict the model capability scores based on these preference results. Nevertheless, as shown in Table 1, the optimization objectives of these methods are often singular, failing to simultaneously achieve the accuracy, convergence, and scalability well. We will discuss this in detail in §2 and conduct experimental validation in §5.2.

Table 1: Optimization Objectives of widely applied CBE Methods. The number of '+' indicates the degree of optimization for the objective, which is discussed in §2 and measured in Table 2.

| Methods | Qin et al. (2024) RANDOM | Chiang et al. (2024) ARENA | Dubois et al. (2024) ALPACAEVAL | Ours UNICBE |
|---|---|---|---|---|
| Accuracy | + | - | - | ++ |
| Convergence | - | + | - | ++ |
| Scalability | - | - | ++ | ++ |

To develop a method that can accurately assess model performance, quickly converge evaluation results, and ensure good scalability when new models are introduced, we theoretically analyze and summarize the following guidelines:

- Improving the *accuracy* of evaluation results relies on completely **uniform** sampling of tuple combinations, so as to mitigate sampling bias.
- Accelerating the *convergence* process involves ensuring the **uniformity** of the win rate uncertainty matrix during its descending process to reduce observation variance.
- Enhancing *scalability* requires sufficient budgets being allocated to new added models to ensure the **uniform** allocation among models, which helps reduce updating uncertainty.

Based on these insights, we propose UNICBE, a unified uniformity-driven framework that can achieve CBE with better accuracy, convergence and scalability. In each iteration of the evaluation process, we first establish sampling probability matrices under different optimization objectives respectively based on real-time preference results. Afterwards, we integrate these matrices to obtain a global sampling probability matrix. Furthermore, we explore various tuple sampling strategies and preference aggregation methods to achieve optimal evaluation results.

To comprehensively validate the effectiveness and generalizability of UNICBE, we conduct experiments involving various types of judges (LLMs and humans), different benchmarks, varied model sets to be evaluated, diverse scenarios (static and dynamic), and multiple evaluation metrics. The main results indicate that, compared to random sampling baseline, UNICBE saves over 17% of evaluation budgets when achieving the same assessment accuracy (with a Pearson coefficient exceeding 0.995 with the ground truth), demonstrating significantly better convergence and accuracy than baselines. Furthermore, in scenarios where new models are continuously introduced, UNICBE save over 50% of evaluation costs compared to random sampling, showcasing excellent scalability.

## 2 RELATED WORK

Comparative preference signals have long been used for model training (Ouyang et al., 2022; Touvron et al., 2023) and evaluation (Chiang et al., 2024; Yuan et al., 2024). Centered around comparing-based evaluation, we will discuss existing budget allocation strategies and preference aggregation methods below.

**Budget Allocation** Many efforts have been made to explore preference budget allocation approaches. The most naive method is to randomly select (models, sample) tuple for judging each time until the preference budget is reached (Qin et al., 2024). This method ensures a relatively uniform sampling across tuple combinations in expectation, thereby guaranteeing the accuracy of evaluation results according to derivation in §3.2. Arena (Chiang et al., 2024) aims to sample model pairs proportionally to the variance gradient of win rate at each step, seeking to accelerating the convergence of evaluation by reducing the uncertainty of the observed win rate matrix in a greedy manner. AlpacaEval (Dubois et al., 2024) measures model performance by comparing the models under evaluation with a fixed reference model. When new models are introduced, preference budget is prioritized for them to stabilize the estimation of their capabilities, thereby achieving good scalability. Despite these methods performing well in their intended objectives, they cannot achieve a balance among accuracy, convergence, and scalability. This makes it imperative to explore better preference budget allocation strategy that can effectively reconcile all these attributes.

**Preference Aggregation** Due to the possibility that the same group of models may exhibit different ranking relationships across different samples, it is essential to estimate the global model capability scores to better fit these non-transitive preference results. Dubois et al. (2024); Zheng et al. (2023) directly use the average pair-wise win rate of each model as a measure of its capability. Feng et al. (2024); Wu & Aji (2023) apply the classical Elo rating system (Elo & Sloan, 1978) (see the Appendix B.1 for detailed introduction) by treating the evaluation process as a sequence of model battles in order to derive model scores. Fageot et al. (2024); Chiang et al. (2024) employ the Bradley-Terry model (Bradley & Terry, 1952) (see Appendix B.2 for detailed introduction) to estimate model scores by maximizing the likelihood of the comparison results between models. We will systematically compare the effectiveness of these preference aggregation methods in §5.3.

## 3 PRELIMINARY

In this section, we start by symbolically introducing the working process of CBE. Afterwards, we introduce the key objectives for achieving efficient CBE: accuracy, convergence, and scalability, and analyze the factors that influence them. We mainly discuss the pair-wise evaluation scenario (where the judge provides preference between two models per time) for its wide applications (Tashu & Horváth, 2018; Qin et al., 2024). Actually, list-wise preferences can be easily converted into pair-wise ones, as demonstrated in §5.4, so the discussions below are general for CBE.

### 3.1 PROCESS OF CBE

Generally, a CBE method $f$ can be divided into three parts: budget allocation strategy $f^{ba}$, tuple sampling strategy $f^{ts}$, and preference aggregation strategy $f^{pa}$. Given benchmark $\mathcal{D} : s_{1:N}$ and models under evaluation $\mathcal{M} : m_{1:M}$, we iterate the following steps: *step 1.* applying $f^{ba}$ to attain sampling matrix $P^l$ at iteration $l$, where $P^l_{i,j,k}$ denotes the probability to select tuple $(m_i, m_j, s_k)$ for judging; *step 2.* applying $f^{ts}$ to sample certain tuple $(m^{l1}, m^{l2}, s^l)$ based on $P^l$; *step 3.* attaining preference result $r^l$ from the judge, where $r^l \in [0, 1]$ denotes the degree $m^{l1}$ wins over $m^{l2}$ (0.5 means tie). We stop this iterative process when the preset preference budget $T$ is achieved and then apply $f^{pa}$ on preference results $\{(m^{l1}, m^{l2}, s^l, r^l)\}^T_{l=1}$ to attain estimated model scores $u_{1:M}$.

### 3.2 ACCURACY

Theoretically, if we have a budget of $\hat{T} = \frac{NM(M-1)}{2}$, we can explore all tuples to obtain the ground truth estimation for the model scores $\hat{u}_{1:M}$. However, typically $T$ is much smaller than $\hat{T}$ in reality considering the preciousness of preference signals. Previous studies (Vabalas et al., 2019; Kossen et al., 2021) have discussed the risks of introducing sampling bias in incomplete sampling scenarios, which we believe could similarly lead to potential risks in CBE. Considering that the content of each sample is $(m^{l1}, m^{l2}, s^l)$, we think the sample bias exists across both samples and models.

**Bias across Samples.** Since different models may excel at answering different types of queries, the model scores can vary depending on the sampled data:

$$u_t = f^{pa}(\{(m_i, m_j, s_k, r_{i,j,k})\}_{i \in 1:M, j \in i+1:M})_t = \hat{u}_t + \eta_{m_t,-,s_k} \qquad \text{for} \quad \forall t, k \quad (1)$$

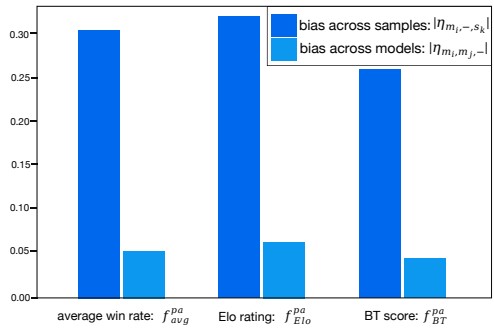
(a) Sampling bias with different preference aggregation strategies across samples and models.

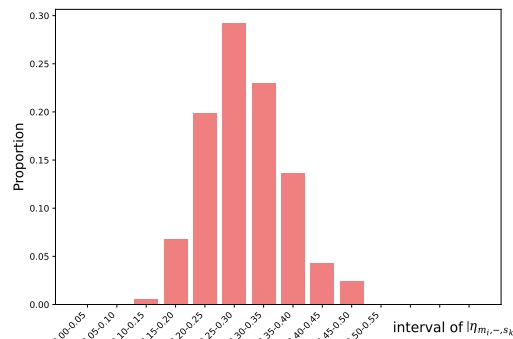
(b) Interval distribution of bias across samples with $f_{BT}^{pa}$ as preference aggregation strategy.

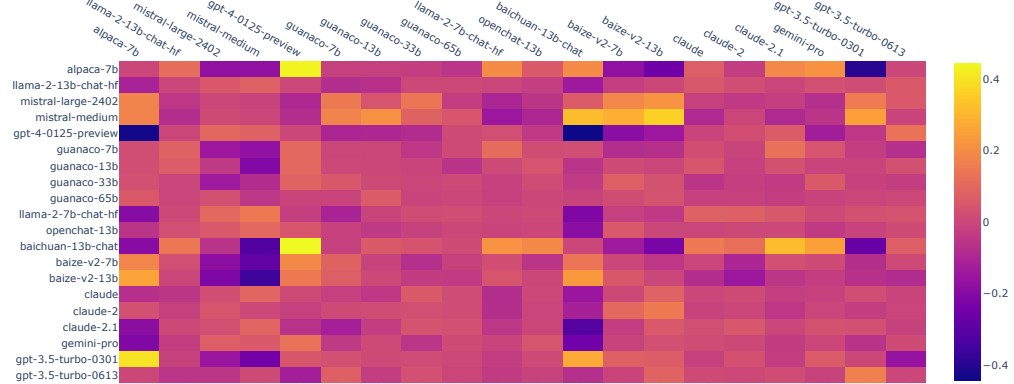
(c) Bias across models with $f_{BT}^{pa}$ as preference aggregation strategy.

Figure 2: Analyses of potential sampling bias risks in CBE.

where $\eta_{m_t,\cdot,s_k}$ represents the bias between the observed model score $u_t$ of $m_t$ and the ground truth $\hat{u}_t$ when sorely assessing on sample $s_k$. To verify this, we conduct experiments on the AlpacaEval benchmark (Dubois et al., 2024) using GPT-4o (OpenAI, 2024) as the judge across randomly selected 20 LLMs (listed in Figure 2(c)). We first traversed all model pairs for samples $s_{1:N}$ to obtain corresponding $N$ sets of preference results and then calculate the respective $|\eta_{m_i,\cdot,s_k}|$ for $i \in 1 : M$ and $k \in 1 : N$ according to equation 1 (model scores are normalized to an average of 1). We calculate the average value of $|\eta_{m_i,\cdot,s_k}|$ across models and samples using different preference aggregation strategies $f^{pa}$ discussed in §2. As shown in Figure 2(a), with all kinds of $f^{pa}$, the average difference between the model scores estimated on single sample and the ground truth values exceeds 0.25, indicating a significant bias across samples. We further analyze the proportion of samples with different biases using $f_{BT}^{pa}$ in Figure 2(b) and find that they overall follow a Gaussian distribution, showing the wide existence of sample bias in CBE.

**Bias across Models.** Just as humans may perform differently when facing different opponents, models may also have varying scores when competing against different models:

$$u_i = f^{pa}(\{(m_i, m_j, s_k, r_{i,j,k})\}_{k \in 1:N})_i = \hat{u}_i + \eta_{m_i,m_j,\cdot} \qquad \text{for } \forall i, j \qquad (2)$$

We validate this from two perspectives: (1) We calculate the average $|\eta_{m_i,m_j,\cdot}|$ according to equation 2 like the process above and show the results in Figure 2(a). Overall, although the bias across models is significantly lower than the bias across samples, it still exists at a scale around 0.05. We further visualize the pair-wise model score bias in Figure 2(c) to validate its wide existence. (2) We obtain over 1.7 million pairwise preference results across 129 LLMs collected by Chatbot Arena [*]. After excluding pairs with fewer than 50 comparisons, we calculate the pairwise win rates and find non-transitivity in 81 model triplets (win rate: $A > B, B > C, C > A$), which also verifies the existence of bias across models.

---

[*] https://storage.googleapis.com/arena_external_data/public/clean_battle_20240814_public.json

**Uniform Allocation Brings the Least Bias.** Based on the discussions above, we analyze the budget allocation strategy that can introduce the least bias. Considering the presence of sampling bias, the estimation error of $u_i$ with $T$ evaluation budget can be expressed as follows:

$$u_i - \hat{u}_i = \sum_{l=1}^{T} \mathbf{1}_{m^{l1}=m_i} \times \eta_{m^{l1},m^{l2},s^l} \tag{3}$$

Considering that $u = \hat{u}$ when all the tuples are traversed, we have the following equation:

$$0 = u_i - \hat{u}_i = \sum_{j=1}^{M} \sum_{k=1}^{N} \eta_{i,j,k} \qquad \text{for} \quad \forall \, i \tag{4}$$

The goal of obtaining the minimum estimation error for $u_i$ is transformed into sampling $T$ numbers ( equation 3) from $MN$ numbers that sum to zero ( equation 4), such that the absolute value of the sum of these $T$ numbers is minimized. We have provided a detailed proof in Appendix A that the best strategy is completely uniform sampling. ***This denotes that the score estimation error can be minimized when the preference budgets are uniformly distributed across models and samples to bring the least sampling bias.***

### 3.3 CONVERGENCE

During the evaluation process, as new preference results are continuously observed, the estimated values of the models win rate matrix and model scores also change constantly. To accelerate the convergence process, we analyze the uncertainty of the win rate matrix as follows. Defining that:

$$X_{i,j}^{l} = \frac{1}{P_{i,j}^{l}} r^l \mathbf{1}_{m^{l1}=m_i \ \& \ m^{l2}=m_j} + \frac{1}{P_{j,i}^{l}} (1 - r^l) \mathbf{1}_{m^{l1}=m_j \ \& \ m^{l2}=m_i} \tag{5}$$

The unbiased estimated win rate matrix $\Phi$ at iteration $L$ can be calculated as follows:

$$\Phi^L = \frac{1}{L} \sum_{l=1}^{L} X^l \tag{6}$$

We further estimate the variance matrix $\Theta$ as:

$$\Theta^L = \frac{1}{L} \sum_{l=1}^{L} (X^l - \Phi^L) \circ (X^l - \Phi^L) \tag{7}$$

Denoting if the model pair $(m_i, m_j)$ has been compared on sample $s_k$ after $l$ iterations as $C_{i,j,k}^{l}$, the uncertainty (standard deviation) of each element in the win rate matrix is as follows:

$$\epsilon_{i,j}^{l} = \sqrt{\frac{\Theta_{i,j}^{l}}{\sum_{k=1}^{N} C_{i,j,k}^{l}}} \tag{8}$$

Allocating the next preference budget on $(m_i, m_j)$ can reduce the uncertainty of their win rate by:

$$\sqrt{\frac{\Theta_{i,j}^{l}}{\sum_{k=1}^{N} C_{i,j,k}^{l}}} - \sqrt{\frac{\Theta_{i,j}^{l}}{\sum_{k=1}^{N} C_{i,j,k}^{l} + 1}} \tag{9}$$

Considering that our core objective is to conduct accurate capability assessments for all models and estimate their ranking relationship, ***we should globally ensure the uniformity of the win rate uncertainty matrix during its descending process to achieve smooth evaluation convergence***.

### 3.4 SCALABILITY

Due to the continuous emergence of new LLMs, the demand for scalability in evaluation method is becoming increasingly prominent (Chern et al., 2024). Considering that we have evaluated $m_{1:M}$ with $T$ budgets, when model $m_{M+1}$ is introduced for assessment, a well-scalable CBE method

should be able to quickly calibrate the capability estimates of $m_{1:M+1}$ with minimal additional preference budget. In this scenario, at the beginning stage when $m_{M+1}$ is introduced, $\text{avg}(C_{M+1,\cdot,\cdot})$ is much smaller than $\text{avg}(C_{\neq M+1,\cdot,\cdot})$. According to equation 8, the uncertainty at this point mainly arises from $\epsilon_{M+1}$, which is also intuitively easy to understand. ***Therefore, the key to improving scalability lies in allocating sufficient evaluation budgets to the newly added models to ensure the uniform allocation among models, reducing the updating uncertainty.***

## 4 UniCBE

The discussions above reveal guidelines for strengthening scalability, accuracy, and convergence in CBE. Based on this, we propose UniCBE, a unified uniformity-driven framework that can simultaneously enhance these objectives well.

### 4.1 Budget Allocation

To ensure the uniformity of tuple combination sampling for minimizing the introduction of sampling bias according to §3.2, we construct $P^{acc\text{-}l}$ at iteration $l$ as follows:

$$P_{i,j,k}^{acc\text{-}l} = \alpha^{-\sum_{k=1}^{N} C_{i,j,k}^{l}} \times \alpha^{-\sum_{i=1}^{M} C_{i,j,k}^{l}} \times \alpha^{-\sum_{j=1}^{M} C_{i,j,k}^{l}} \tag{10}$$

where $\sum_{k=1}^{N} C_{i,j,k}^{l}$ denotes the times model pair $(m_i, m_j)$ has been compared, $\sum_{i=1}^{M} C_{i,j,k}^{l}$ and $\sum_{j=1}^{M} C_{i,j,k}^{l}$ denote the times model $m_i$ and $m_j$ has been tested on $s_k$ respectively. If certain model-model combination or model-sample combination have been sampled multiple times, equation 10 will reduce the probability of such combinations being selected again, thereby achieving sufficient uniformity to minimize the introduction of bias between models and samples, respectively.

To accelerate the convergence of evaluation results, we construct $P^{con\text{-}l}$ according to §3.3 as follows:

$$P_{i,j,k}^{con\text{-}l} = \epsilon_{i,j}^{l} \tag{11}$$

Sampling specific model pair helps reduce the uncertainty of their win rate estimation according to equation 9. By sampling proportionally to the win rate uncertainty matrix, we can uniformly decrease the uncertainty for each model pair, thereby facilitating convergence.

We construct $P^{sca\text{-}l}$ to allocate more preference budget to the newly introduced model so as to improving the scalability according to §3.4 as follows:

$$P_{i,j,k}^{sca\text{-}l} = \alpha^{-\sum_{k=1}^{N} \sum_{i=1}^{M} C_{i,j,k}^{l}} \times \alpha^{-\sum_{k=1}^{N} \sum_{j=1}^{M} C_{i,j,k}^{l}} \tag{12}$$

Finally, we integrate the matrices mentioned above to obtain $P^l$, ensuring that sampling according to $P^l$ can simultaneously balance the accuracy, convergence, and scalability of evaluation results:

$$P^l = \frac{P^{acc\text{-}l} \circ P^{con\text{-}l} \circ P^{sca\text{-}l}}{\sum(P^{acc\text{-}l} \circ P^{con\text{-}l} \circ P^{sca\text{-}l})} \tag{13}$$

### 4.2 Tuple Sampling

After obtaining $P^l$, we need to sample tuples for judging based on it. Two tuple sampling strategies are considered:

- **probabilistic sampling** $f_p^{ts}$ means sampling tuple directly according to $P^l$.
- **greedy sampling** $f_g^{ts}$ means selecting the tuple with the maximum probability in $P^l$.

The default tuple sampling strategy of UniCBE is $f_g^{ts}$, which can avoid the suboptimal achievement of objectives due to uncertainty in the sampling process.

### 4.3 Preference Aggregation

As discussed in §2, mainstream preference aggregation strategies include averaging win rate $f_{avg}^{pa}$, Elo rating system $f_{Elo}^{pa}$ and Bradley-Terry model $f_{BT}^{pa}$. In our preliminary experiment (Figure 2(c)) we observe that $f_{BT}^{pa}$ can better alleviate sampling bias, for which we choose it as our default setting.

## 5 EXPERIMENTS

Centered around UNICBE, we will empirically compare its performance with baselines and validate its scalability in §5.2, explore the optimal variants in §5.3, and demonstrate its generalizability under different settings in §5.4.

### 5.1 EXPERIMENTAL SETTINGS

**Benchmarks.** We choose AlpacaEval (Dubois et al., 2024) and MT-Bench (Zheng et al., 2023) benchmarks for our experiments. For AlpacaEval, we use its default version which includes 805 high-quality human annotated instructions and corresponding responses from multiple LLMs. We randomly choose 20 LLMs (listed in Figure 2(c)) for experiments, with GPT-4o and GPT-3.5-turbo as judges (see Appendix D for the prompt). For MT-Bench, we use the released responses from the all 6 LLMs and corresponding human preferences for experiments.

**Baselines.** We choose widely applied [†] methods RANDOM, ARENA and ALPACAEVAL as baselines, which have been discussed in §2 and listed in Table 1.

**Metrics.** To assess the effectiveness of the CBE methods, we evaluate the accuracy of both the estimated model pair-wise win rates and the model scores. We calculate the average absolute error between the estimated win rates and corresponding ground truth (the estimates when $T = \hat{T}$). We calculate the Spearman correlation coefficient $r_s$ between the predicted model scores and the corresponding ground truth to evaluate the accuracy of the model's rank-order relationship, and the Pearson correlation coefficient $r_p$ to assess the accuracy of the linear relationship.

**Details.** To ensure the reliability of the experimental results, for each setting, we randomly select $M$ (default to 15 for AlpacaEval and 5 for MT-Bench) models and $N$ (default to 805 for AlpacaEval and 700 for MT-Bench) samples, and report the average results across 10,000 random seeds. We don't observe obvious performance difference in preliminary experiments when varying $\alpha$ within the range of $[1.5, 3]$ (we conduct a detailed discussion about this in Appendix F.1), thus we set the default value of $\alpha$ as 2 in our experiments.

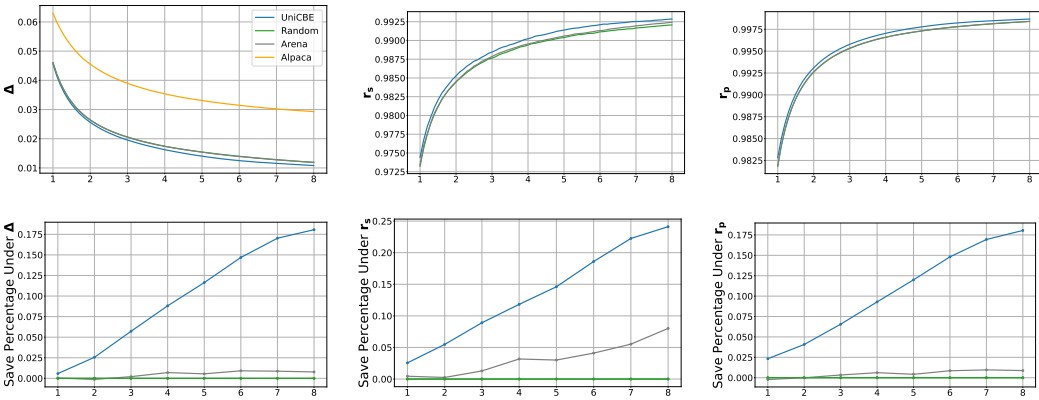

Figure 3: Results of compared CBE methods with GPT-4o as the judge on AlpacaEval benchmark. The X-axis (applicable to all plots below) represents the preference budget ($k$). $\mathbf{\Delta}$ denotes the mean absolute error of the estimated win rate. $\mathbf{r_s}$ and $\mathbf{r_p}$ denote the Spearman and Pearson correlations between the the estimated model scores and the ground truth respectively.

### 5.2 MAIN RESULTS

**Accuracy and Convergence.** The results of compared CBE methods on AlpacaEval benchmark with GPT4-turbo as the judge are shown in Figure 3. To better illustrate the results, we also calculate the percentage of preference budget saved by each method compared to RANDOM baseline when

---

[†] https://tatsu-lab.github.io/alpaca_eval/, https://lmarena.ai/

achieving the same performance. In terms of performance, ALPACAEVAL $<<$ RANDOM $<$ ARENA $<$ UNICBE. To understand the differences in the performance of each method, we quantitatively analyze them based on the guidelines summarized in § 3. To achieve accuracy, convergence, and scalability, it is supposed to allocate the preference budget in a way that ensures uniformity across tuples, uniformity across model pairs in win-rate uncertainty, and uniformity across models. We calculate the cosine similarity between the allocation results of these methods and the corresponding expected uniform vectors for each objective as a measure, denoted as $\beta_{acc}$, $\beta_{con}$, and $\beta_{sca}$, respectively (see Appendix E for calculating process). As shown in Table 2, the fixed inclusion of the reference model in the tuple selection of ALPACAEVAL compromises uniformity across multiple aspects, thereby resulting in lower $\beta$ values and significantly poorer performance. ARENA and RANDOM respectively improve the balance of uncertainty and suppression of sampling bias, resulting in higher $\beta_{con}$ and $\beta_{acc}$ values. Following our guidelines, UNICBE improves $\beta_{con}$, $\beta_{acc}$, and $\beta_{sca}$ simultaneously and save over 17% of the preference budget compared to RANDOM with a $\Delta$ close to 0.01, showcasing improved accuracy and convergence.

Table 2: The measurement results of the achievement of objectives in §3 for the compared methods.

| Methods | RANDOM | ARENA | ALPACAEVAL | UNICBE |
|---|---|---|---|---|
| $\beta_{acc}$ | .5803 | .5725 | .0925 | .7364 |
| $\beta_{con}$ | .9081 | .9172 | .3515 | .9228 |
| $\beta_{sca}$ | .9972 | .9945 | .4987 | .9997 |

**Scalability.** To analyze scalability, we establish a scenario where we initially have 11 models awaiting assessment, and new models are sequentially added every 2000 samplings. As shown in Figure 4, Whenever a new model is introduced, UNICBE can rapidly stabilize the performance through adaptive preference allocation skewing for the new model, saving over 50% of the budget compared to the RANDOM baseline. In contrast, ARENA and RANDOM exhibit poorer scalability since they do not consider scalability as optimization objective. Although the budget allocated to the reference model is significantly more than that for other models, resulting in a lower $\beta_{sca}$ for ALPACAEVAL, the strategy of automatically allocating the budget to the new introduced models also provides it with good scalability.

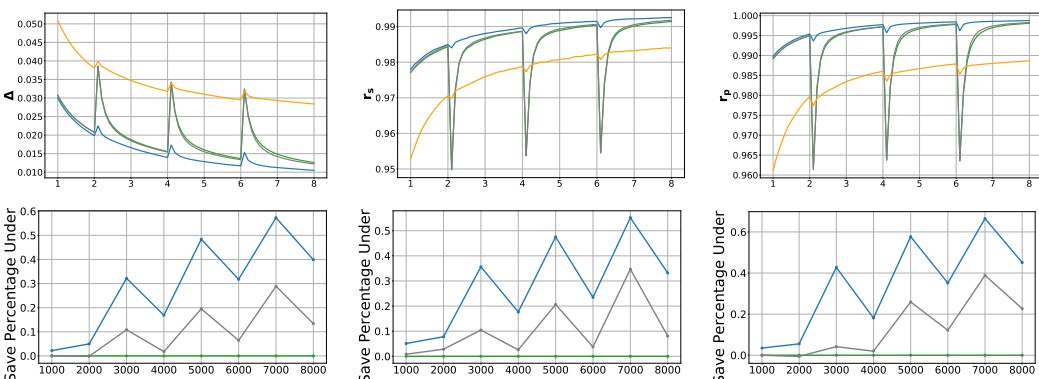

Figure 4: Results of compared CBE methods in the scenario where new model are consistently introduced every 2000 iterations.

## 5.3 VARIANTS ABLATIONS

**Budget Allocation Objectives.** We test the impact of different optimization objectives by removing $P^{acc}$, $P^{con}$, and $P^{sca}$ from equation 13 separately. As shown in Figure 5, the significant performance degradation observed when removing $P^{acc}$ from UNICBE indicates that mitigating sampling bias to improve accuracy is the most critical factor in achieving efficient CBE. Furthermore, we find that $P^{con}$ has a considerable impact on $r_s$. We hypothesize that this is because balancing the uncertainty among different models helps prevent any one model from having a significant ranking

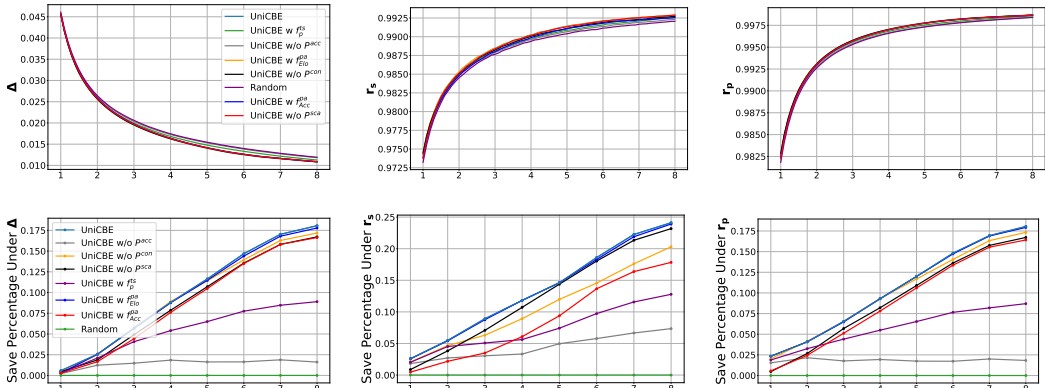

Figure 5: Ablation studies of UNICBE with GPT-4o as the judge on AlpacaEval benchmark.

bias due to its larger uncertainty. The performance drop when removing $P^{sca}$ also suggests that ensuring uniformity in sampling across models not only enhances scalability but also further reduces sampling bias, thereby improving accuracy.

**Tuple Sampling and Preference Aggregation Strategies.** As shown in Figure 5, replacing greedy sampling with probabilistic sampling $f_p^{ts}$ results in a significant performance drop. This is likely because the randomness introduced by $f_p^{ts}$ hinders the achievement of multiple optimization objectives. In terms of preference aggregation strategies, the Elo rating system $f_{Elo}^{pa}$ shows a slight performance decline compared to the BT model due to its higher instability (Boubdir et al., 2023). Moreover, the strategy of directly using the average win rate $f_{avg}^{pa}$ may introduce additional bias, as it fails to consider the varying strengths of the opponents faced by different models, leading to a performance decrease.

## 5.4 GENERALIZABILITY UNDER DIFFERENT SETTINGS

**Different Judges.** Apart from GPT-4o, we also experiment with GPT-3.5-turbo and Qwen-Plus as judge on AlpacaEval, and human as judge on MT-Bench. As shown in the above part of Figure 6, the overall conclusion with GPT-3.5-turbo is similar to GPT-4o, except for: (1) The performance of ARENA no longer shows advantage over RANDOM. (2) There is a certain decline in the performance of all methods, which is likely due to the increased noise in the preferences provided by GPT-3.5-turbo, leading to slower convergence. Similar trends are observed with Qwen-Plus (See Figure 14). Results on MT-Bench are shown in the below part of Figure 6, where UNICBE also demonstrates better performance compared to other methods. However, due to the limited preference data included in MT-Bench, the experimental results show relatively larger fluctuations. The results above demonstrate the good generalizability of UNICBE across different judges and the data domain.

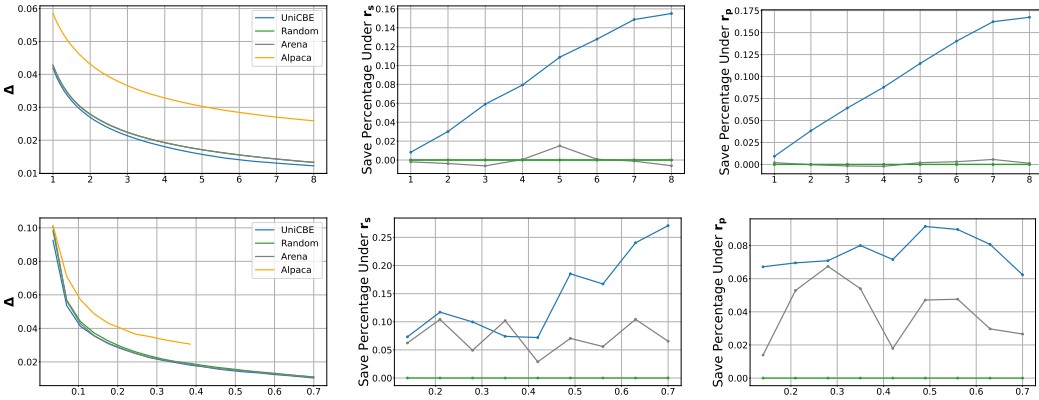

Figure 6: Results of compared CBE methods with GPT-3.5-turbo as the judge on AlpacaEval (above) and human as the judge on MT-Bench (below).

**Varied Number of Models and Samples.** Finally, as shown in Figure 7, we conducte experiments by varying the number of models $M$ and samples $N$. It can be observed that UNICBE achieves significantly better results compared to all the baselines under these settings, especially when $M$ and $N$ are large.

**List-wise Preference.** UNICBE can also be applied to list-wise preference. Suppose the judge provides a preference ranking for $K$ models each time. We can compute a $K + 1$ dimensional $P$, similar to equation equation 13, and sample to obtain a tuple $(m^{l\text{-}1}, ..., m^{l\text{-}K}, s^l)$. From the judge's ranking of this tuple, we derive $\frac{K(K-1)}{2}$ pair-wise preferences. Figure 8 shows the results for the case where $K = 3$. It can be seen that UNICBE achieves a savings compared to RANDOM of over 30% in this setting. This may be due to the fact that list-wise preference results in $\frac{K(K-1)}{2}$ pair-wise preferences concentrated among the $K$ models, exacerbating the sampling bias. Therefore, UNICBE is even more needed to suppress this effect with list-wise preference.

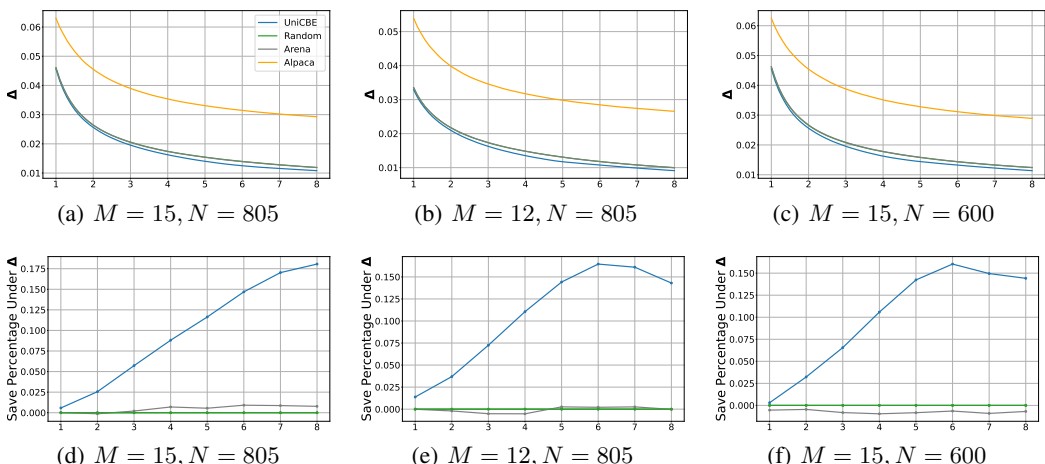

Figure 7: Effects of number of models under evaluation $M$ and number of samples $N$. The results are obtained with GPT-4o as the judge on AlpacaEval.

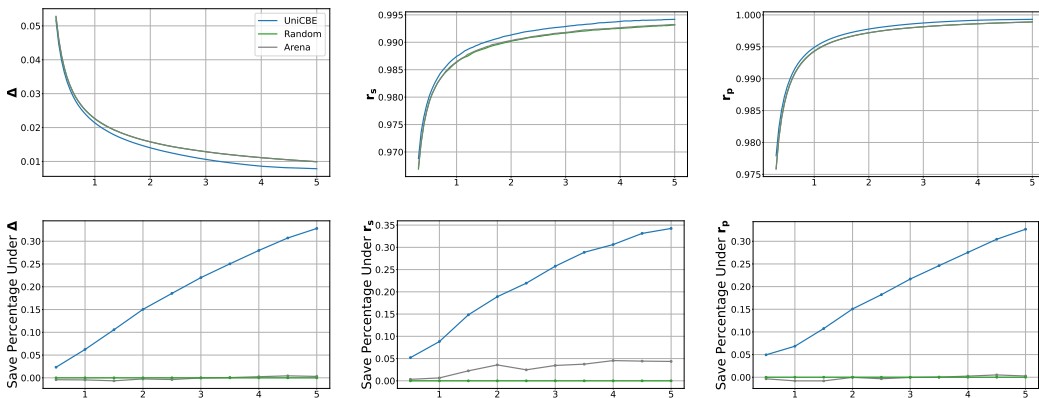

Figure 8: Performance of CBE methods with list-wise preference of GPT-4o on AlpacaEval.

## 6 CONCLUSIONS

The existing comparing-based evaluation methods are ineffective in fully utilizing valuable preference signals due to their constrained optimization objectives. Our in-depth analysis reveals that the key to enhancing CBE lies in mitigating sampling bias, balancing the descent process of uncertainty, and suppressing the updating uncertainty. Based on this, we propose the UNICBE framework that simultaneously optimizes the aforementioned objectives by promoting uniformity in corresponding aspects to enhance accuracy, convergence, and scalability. Comprehensive experiments and analyses confirm the strong effectiveness, improved scalability, and good generalizability of UNICBE.

ACKNOWLEDGEMENTS

This work is supported by the Beijing Natural Science Foundation, China (Nos. 4222037, L181010).

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

## A  PROOF OF THEOREM IN §3.2

Given that $\sum_{i=1}^{U} X_i = 0$, we want to attain sampling set $\mathcal{S}$ that satisfies $|\mathcal{S}| = V$ and:

$$\mathcal{S} = argmin_{\mathcal{S}}|\sum_{i\in\mathcal{S}} X_i| = argmin_{\mathcal{S}}(\sum_{i\in\mathcal{S}} X_i)^2 \tag{14}$$

Firstly, it is easy to know that for any sampling set $\mathcal{S}$:

$$\mathbb{E}[\sum_{i\in\mathcal{S}} X_i] = 0 \tag{15}$$

Thus,

$$\mathbb{E}[(\sum_{i\in\mathcal{S}} X_i)^2] = \mathbb{E}[(\sum_{i\in\mathcal{S}} X_i - 0)^2] = \mathbb{E}[(\sum_{i\in\mathcal{S}} X_i - \mathbb{E}[\sum_{i\in\mathcal{S}} X_i])^2] = \mathbb{E}[\mathrm{Var}[\sum_{i\in\mathcal{S}} X_i]] \tag{16}$$

Considering that:

$$\mathrm{Var}[\sum_{i\in\mathcal{S}} X_i] = \sum_{i\in\mathrm{set}(\mathcal{S})} c_i^2 \mathrm{Var}(X_i) \tag{17}$$

where $c_i$ denotes the number of $X_i$ in $\mathcal{S}$. On this basis, we derive that:

$$
\begin{aligned}
\mathbb{E}[(\sum_{i\in\mathcal{S}} X_i)^2] &= \mathbb{E}[\mathrm{Var}(X)] \sum_{i\in\mathrm{set}(\mathcal{S})} c_i^2 \\
&\geq \mathbb{E}[\mathrm{Var}(X)](\sum_{i\in\mathrm{set}(\mathcal{S})} c_i)^2 |\mathrm{set}(\mathcal{S})|^{-1} \\
&= V^2 \mathbb{E}[\mathrm{Var}(X)]|\mathrm{set}(\mathcal{S})|^{-1} \\
&\geq V^2 \mathbb{E}[\mathrm{Var}(X)]\min(U, V)^{-1}
\end{aligned} \tag{18}
$$

The equality condition of the first inequality is: the number of samples taken from each category is equal. The equality condition of the second inequality is: the number of sampled categories equals to $\min(U, V)$. These two conditions imply that a completely uniform sampling strategy is optimal.

## B  INTRODUCTION OF ELO RATING SYSTEM AND BRADLEY-TERRY MODEL

### B.1  ELO RATING SYSTEM

The Elo rating system (Elo & Sloan, 1978) is widely used to rank participants based on their relative performance in competitive settings. Given two models, $A$ and $B$, with initial ratings $R_A$ and $R_B$, the expected score of model $A$ in a pairwise comparison is calculated as:

$$E_A = \frac{1}{1 + 10^{(R_B - R_A)/400}}$$

Similarly, the expected score for model $B$ is:

$$E_B = \frac{1}{1 + 10^{(R_A - R_B)/400}}$$

After the comparison, the actual results are used to update the ratings. If model $A$ wins, its new rating $R'_A$ is updated as:

$$R'_A = R_A + K(S_A - E_A)$$

where $S_A$ is the actual result of the match (1 for a win, 0 for a loss, and 0.5 for a draw), and $K$ is a constant that controls the sensitivity of the rating adjustment. Model $B$'s rating is updated in a similar way:

$$R'_B = R_B + K(S_B - E_B)$$

where $S_B$ is the actual result for model $B$.

When extending the Elo rating system to multiple models, we consider a set of $n$ models. Pairwise comparisons between the models are conducted, resulting in $\binom{n}{2}$ unique pairs:

$$\binom{n}{2} = \frac{n(n-1)}{2}$$

Each pair is evaluated using the Elo score update rules, and the results are iteratively applied to adjust the ratings, ensuring that each model's rating reflects its relative performance within the set.

The extension to multiple models leverages the transitive property. For any three models $i, j, k \in \{1, 2, \ldots, n\}$, if $R_i > R_j$ and $R_j > R_k$, the transitivity implies:

$$R_i > R_j \quad \text{and} \quad R_j > R_k \quad \implies \quad R_i > R_k$$

This property ensures consistency in the rankings, even when individual match outcomes vary. By iterating over all $\binom{n}{2}$ comparisons, the Elo scores converge to reflect the overall capabilities of the models, with higher scores indicating stronger performance.

### B.2 BRADLEY-TERRY MODEL

The Bradley-Terry model (Bradley & Terry, 1952) estimates the probability that one model outperforms another in pairwise comparisons. For two models $M_i$ and $M_j$ with strength parameters $\xi_i$ and $\xi_j$, the probability that model $i$ beats model $j$ is modeled as:

$$P(M_i > M_j) = \frac{1}{1 + e^{\xi_j - \xi_i}} \tag{19}$$

where $\xi$ is an $|M|$-length vector of Bradley-Terry coefficients. Given a set of comparing results $\mathcal{S} = \{(M_i^t, M_j^t, R^t)\}_{t=1}^T$ where $R^t$ represents the degree $M_i^t$ wins over $M_j^t$. We set $\text{mean}(\xi) = 0$. After attaining the BT scores using $f^{pa}$, we calculate the estimated win rate matrix with equation 19.

## C MORE EXPERIMENTAL ANALYSES

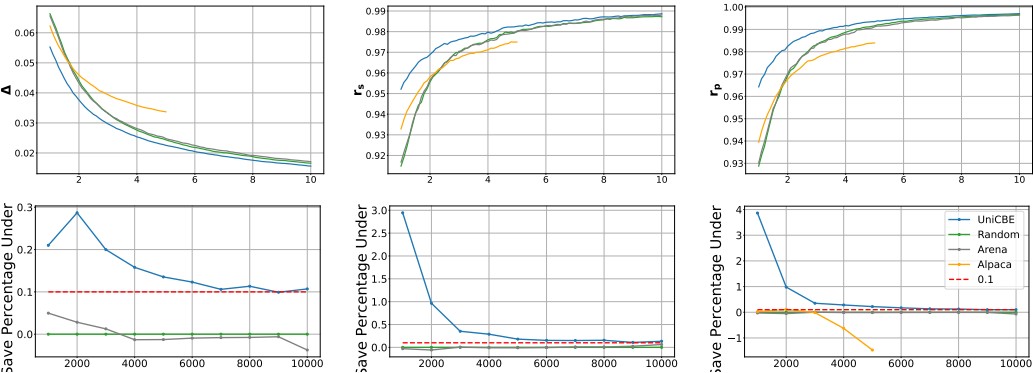

Figure 9: Results of compared CBE methods in a scenario where models and samples are dynamically added or removed at a random frequency.

### C.1 PERFORMANCE IN SCENARIOS CLOSE TO REALITY

We think that conducting experiments in settings that are closer to real-world scenarios (highly dynamic and requiring real-time evaluation) can help us more comprehensively assess UNICBE and the baseline methods. To this end, we perform the following experiments: Starting with a sample size of $N = 600$ and model number of $M = 12$, we execute a random operation at each time step. The operations included: adding one model to be evaluated with a probability of 0.01, removing one model with a probability of 0.01, adding one potential sample with a probability of 0.01, randomly deleting one sample with a probability of 0.01, and taking no action with a probability of 0.96. Based on the experimental results shown in Figure 9, we have the following observations:

- The convergence speed of all baseline methods significantly slowed down compared to Figure 3. None of the baseline methods achieve a Spearman correlation coefficient of 0.96 or a Pearson correlation coefficient of 0.97 by $T = 2000$, highlighting the difficulty of model evaluation in this setting. In contrast, UNICBE achieve rapid convergence, reaching a Spearman coefficient of approximately 0.97 and a Pearson coefficient exceeding 0.98 by $T = 2000$.
- Over the long term, as $T$ increases, UNICBE consistently demonstrates over 10% savings in preference budget across all metrics, even under this challenging setting, showcasing its strong practicality.
- An interesting observation is that ALPACAEVAL exhibits better convergence in the early stages compared to RANDOM and ARENA, supporting our previous conclusions in Table 1. However, as $T$ increases, ALPACAEVAL's lack of accuracy optimization objective leads to its performance being surpassed by RANDOM and ARENA.

## C.2 ABLATION STUDY OF UNIFORMITY CONSTRAINTS

Based on our analysis in §3, the degree to which uniformity is achieved is positively correlated with performance in terms of accuracy, convergence, and scalability. To explore the empirical relationship between the degree of uniformity constraints and the final outcomes, we draw inspiration from the concept of temperature-based control in random sampling. By adjusting the temperature $T$ in the following formula for sampling $f_T^{ts}$, we regulate the extent of uniformity constraints according to $P^l$ in equation 13:

$$f_T^{ts}(i, j, k) = \frac{(P_{i,j,k}^l)^{-T}}{\sum (P^l)^{-T}} \tag{20}$$

As $T$ increases, the uniformity constraints become more relaxed. When $T = 0$, it corresponds to greedy sampling $f_g^{ts}$, which imposes the strictest uniformity constraints. When $T = 1$, it corresponds to probabilistic sampling $f_p^{ts}$, which imposes general uniformity constraints. When $T = +\infty$, it corresponds to random sampling, where no uniformity constraints are applied. Our experimental results are shown in Figure 10. As $T$ increases from 0 to $T = +\infty$, the evaluation results progressively deteriorate. This indicates that adopting greedy sampling to impose the strictest uniformity constraints yields the optimal evaluation performance. This observation also validates the correctness of our conclusions in §3.

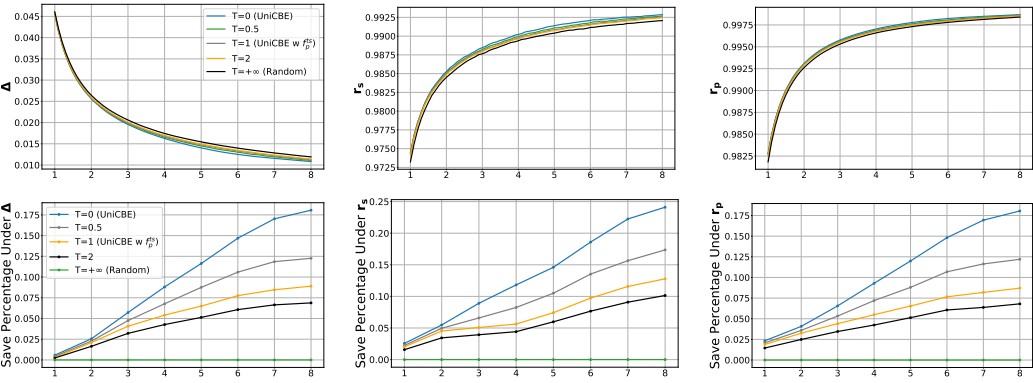

Figure 10: Results of UNICBE with different sampling temperatures.

## C.3 ADJUSTING THE WEIGHTS OF OPTIMIZATION OBJECTIVES

In equation 13, we integrate sampling matrices targeting different optimization objectives with equal weights. In practice, when faced with varying requirements, it is straightforward to prioritize a specific objective by adjusting the weights $\theta_{acc}$, $\theta_{con}$, and $\theta_{sca}$ for these matrices, as shown in equation 21.

$$P^l = \frac{(P^{acc\text{-}l})^{\theta_{acc}} \circ (P^{con\text{-}l})^{\theta_{con}} \circ (P^{sca\text{-}l})^{\theta_{sca}}}{\sum ((P^{acc\text{-}l})^{\theta_{acc}} \circ (P^{con\text{-}l})^{\theta_{con}} \circ (P^{sca\text{-}l})^{\theta_{sca}})} \tag{21}$$

As demonstrated in Table 3, we set different settings and calculate the degree of achievement $\beta$ for each optimization objective following the procedure described in §E. Compared to equal-weight

integration, users can increase the corresponding $\beta$ (e.g., $\beta_{acc}$) by assigning a larger weight to a specific optimization objective ($\theta_{acc}$), thereby better meeting their practical needs (accuracy). We also observe that enhancing a specific optimization objective often comes with a slight decrease in the achievement of other objectives. In Figure 11, we illustrate an example of improving accuracy, where $\theta_{acc}$ is increased from 1 to 2. We find that the increased focus on accuracy objective slightly slows down the convergence speed. As a result, when $T$ is relatively small, the performance of $\theta_{acc} = 2$ lags behind that of $\theta_{acc} = 1$. However, in the later stages, after convergence, the enhanced accuracy objective enables $\theta_{acc} = 2$ to outperform $\theta_{acc} = 1$, resulting in greater savings in the preference budget.

Table 3: The measurement results of the achievement of objectives in §3 for UNICBE with varied hyperparameters.

| Settings | $\theta_{acc} = 2$ $\theta_{con} = 1$ $\theta_{sca} = 1$ | $\theta_{acc} = 1$ $\theta_{con} = 2$ $\theta_{sca} = 1$ | $\theta_{acc} = 1$ $\theta_{con} = 1$ $\theta_{sca} = 2$ | $\theta_{acc} = 1$ $\theta_{con} = 1$ $\theta_{sca} = 1$ |
|---|---|---|---|---|
| $\beta_{acc}$ | .7380(+.0016) | .7355(-.0009) | .7351(-.0013) | **.7364** |
| $\beta_{con}$ | .9221(-.0007) | .9235(+.0007) | .9217(-.0011) | **.9228** |
| $\beta_{sca}$ | .9996(-.0001) | .9997(.0000) | .9998(+.0001) | **.9997** |

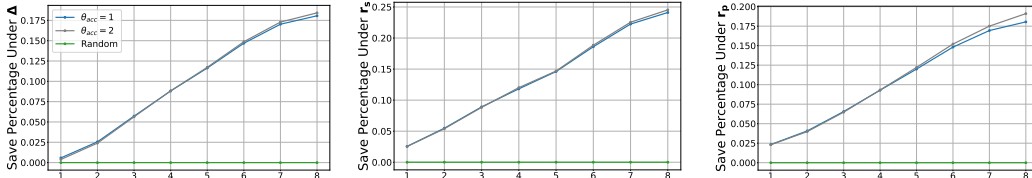

Figure 11: Results of UNICBE with different $\theta_{acc}$.

# D   PROMPT FOR HAVING AN LLM ACT AS JUDGE

We follow AlpacaEval[‡] to instruct the LLMs act as judge with the following prompt:

*You are a helpful assistant, that ranks models by the quality of their answers.*

*I want you to create a leaderboard of different of large-language models. To do so, I will give you the instructions (prompts) given to the models, and the responses of two models. Please give the winner model based on which responses would be preferred by humans. All inputs and outputs should be python dictionaries.*

*Here is the prompt:*
*{*
*"instruction": """{instruction}""",*
*}*

*Here are the outputs of the models:*
*{*
*"model": "model_1",*
*"answer": """{output_1}"""*
*},*
*{*
*"model": "model_2",*

---
[‡]https://github.com/tatsu-lab/alpaca_eval

*"answer": """$\{output_2\}$"""*
*}*

*Now please give the winner model according to the quality of their answers, so that the winner model has the best output. Then return the winner model in the following format if $model_x$ is the winner: winner: $model_x$*

*You need to strictly follow the format above. Please provide the ranking that the majority of humans would give.*

## E   THE CALCULATION PROCESS OF $\beta$ IN TABLE 2

We calculate the $\beta$ values for each CBE method to measure how well they align with the optimization objectives we analyzed in §3: ensuring uniformity across tuples, uniformity across model pairs in win-rate uncertainty, and uniformity across models. Specifically, we first construct $U^{acc1}$, $U^{acc2}$, $U^{con}$ and $U^{sca}$ as follows:

$$U_{i,j}^{acc1} = \begin{cases} 0, & \text{if } i = j \\ \frac{1}{M(M-1)}, & \text{else} \end{cases} \tag{22}$$

$$U_{i,k}^{acc2} = \frac{1}{MN}$$

$$U_{i,j}^{con} = \begin{cases} 0, & \text{if } i = j \\ \frac{1}{M(M-1)}, & \text{else} \end{cases} \tag{23}$$

$$U_i^{sca} = \frac{1}{M} \tag{24}$$

On this basis, we calculate $\beta_{acc}$, $\beta_{con}$ and $\beta_{sca}$ as follows:

$$\beta_{acc} = \text{CosineSim}(\text{U}^{acc1}, \text{C.mean}(\text{dim} = -1)) \times \text{CosineSim}(\text{U}^{acc2}, \text{C.mean}(\text{dim} = 0)) \tag{25}$$

$$\beta_{con} = \text{CosineSim}(\text{U}^{con}, \epsilon) \tag{26}$$

$$\beta_{sca} = \text{CosineSim}(\text{U}^{sca}, \text{C.mean}(\text{dim} = -1).\text{mean}(\text{dim} = -1)) \tag{27}$$

## F   FURTHER DISCUSSIONS

### F.1   AFFINITY OF THE THREE OPTIMIZATION OBJECTIVES

We have discussed in §3 that the keys to strengthen the accuracy, convergence and scalability of CBE are: ensuring uniformity across tuples, uniformity across model pairs in win-rate uncertainty, and uniformity across models. Below we discuss their compatibility. Firstly, ensuring uniformity across models can be seen as a sub-goal of ensuring uniformity across tuples, therefore they exhibit a strong affinity. The former can be considered a refinement of the latter, specifically focusing on model uniformity. Moreover, as shown inequation 8, the uncertainty in win-rate between models $\epsilon_{i,j}$ is inversely proportional to the number of comparisons between models. Therefore, improving uniformity across model pairs in win-rate uncertainty $\epsilon$ will also contribute to a more uniform distribution of comparisons between models. This further implies that ensuring uniformity across model pairs in win-rate uncertainty and ensuring uniformity across tuples are compatible goals with a strong affinity. Furthermore, as shown in Table 2, all $\beta$ values of UNICBE are improved compared to the baselines, experimentally validating the fact that optimizing the three objectives can be mutually beneficial. Based on this analysis, we can infer that changing $\alpha$ essentially adjusts the optimization emphasis on different objectives. However, since the three objectives are mutually reinforcing, the effect of changing $\alpha$ will be relatively small.

### F.2   DISCUSSION ON SAMPLING BIAS IN INCOMPLETE SAMPLING SCENARIOS

Previous studies have discussed the risks of introducing sampling bias in incomplete sampling scenarios. Specifically, Vabalas et al. (2019) demonstrated through simulation experiments that K-fold

cross-validation (K-fold CV) can produce significant performance estimation bias when dealing with small sample sizes. This bias persists even when the sample size reaches 1000. In contrast, methods like nested cross-validation (Nested CV) and train/test split have been shown to provide robust and unbiased performance estimates regardless of sample size. Kossen et al. (2021) introduced a weighting scheme, as described in (Farquhar et al., 2021), to mitigate sampling bias in active testing scenarios. Vivek et al. (2024) proposed leveraging information obtained from source models to select representative samples from the test set, thereby reducing sampling bias. Additionally, Polo et al. (2024) employed Item Response Theory (Lord & Novick, 2008) to correct sample bias in addressing this issue.

These studies inspired us to investigate the bias problem in the CBE scenario. Unlike the aforementioned studies, we found that in CBE scenario, not only does sample bias exist, but model bias also plays a role, and the two are coupled. This coupling poses greater challenges for analyzing and mitigating these biases. To address this, based on the analyses outlined in §3, we propose the UNICBE method, which effectively alleviates biases in this scenario.

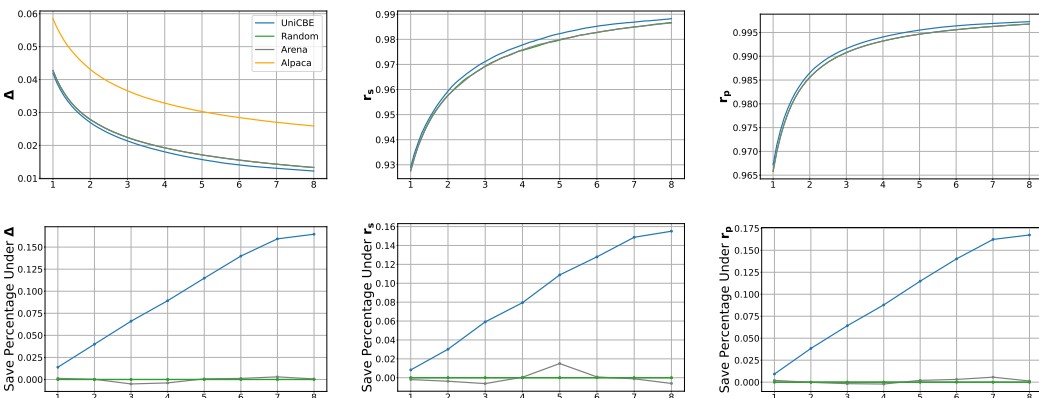

Figure 12: Results of compared CBE methods with GPT-3.5-turbo as the judge on AlpacaEval.

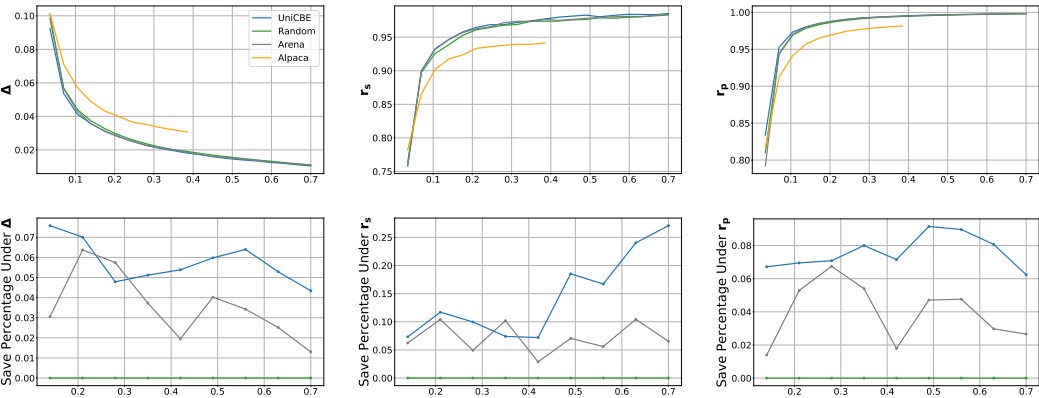

Figure 13: Results of compared CBE methods with human as the judge on MT-Bench.

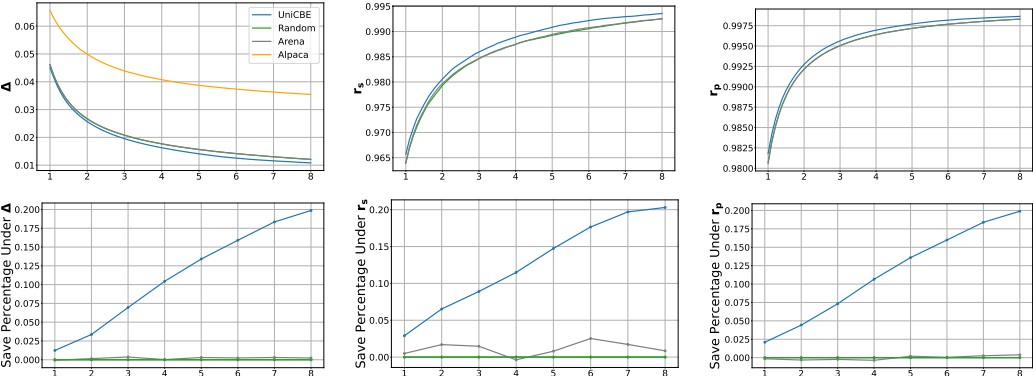

Figure 14: Results of compared CBE methods with Qwen-Plus as the judge on AlpacaEval.

