# OpenReview forum: "UniCBE: An Uniformity-driven Comparing Based Evaluation Framework with Unified Multi-Objective Optimization"
_ICLR.cc/2025/Conference — ICLR 2025 Spotlight_

### Official Review · Reviewer_8sRK · 2024-11-02

**Soundness:** 4
**Presentation:** 2
**Contribution:** 3
**Rating:** 8
**Confidence:** 4

**Summary:**

This paper presents UNICBE, a new framework for comparing-based evaluation (CBE) to better align large language models with human preferences. Unlike traditional CBE methods that focus on single objectives, UNICBE addresses sampling bias, manages uncertainty, and optimizes preference signals through three specialized sampling probability matrices. Tested on the AlpacaEval benchmark, UNICBE achieves high accuracy with a Pearson correlation over 0.995 and reduces evaluation costs by 17%, with savings exceeding 50% when evaluating new models, highlighting its efficiency and scalability.

**Strengths:**

This paper tackle an important question in LLM evaluation and provide a sound solution to it. The experiments are extensive and convincing with strong results showing the advantage of the proposed methods.

**Weaknesses:**

I don't see any major weakness of the paper, just the presentation can be improved, especially lack in explaining why the method is better than others in an intuitive and easy to follow way.

**Questions:**

The authors argue that to avoid bias, the budget should be allocated uniformly, if so how could this method be more sample efficient than random? I guess the reason is if model A is much better than B and model B is much better than C, then it's not necessary to compare A and C a lot. But if thats the reason why this method is more sample-efficient, it would be contradictory to the uniform assumption. Could the authors provide more insight into this?

---

> ### Author Response · Authors · 2024-11-19
> **Response to Reviewer 8sRK**
>
> Dear Reviewer 8sRK, we sincerely thank you for your valuable feedback on our submission. Below is our responses to the concerns you raised. We have incorporated these contents into the updated version of our paper, which we believe will help enhance the quality of our submission.
>
> **We have also provided a PDF version via an anonymous link for a clearer and more complete presentation of our response: https://anonymous.4open.science/r/Respones-to-Reviewer-8sRK-8A44/response%20to%20reviewer%208sRK.pdf. This can also be find in the Supplementary Material.**
>
> > **The presentation can be improved, especially lack in explaining why the method is better than others in an intuitive and easy to follow way. The authors argue that to avoid bias, the budget should be allocated uniformly, if so how could this method be more sample efficient than random? I guess the reason is if model A is much better than B and model B is much better than C, then it's not necessary to compare A and C a lot. But if thats the reason why this method is more sample-efficient, it would be contradictory to the uniform assumption. Could the authors provide more insight into this?**
>
> Your question raises important points for discussion, and we will respond from two perspectives:
>
> First, you mentioned leveraging the transitivity of model performance to reduce the preference budget. This approach is feasible when the optimization goal is to determine the relative ranking of models or to identify the best model (as demonstrated by the UCB algorithm [1]). However, our goal is to precisely evaluate the true capability values of each model rather than their order or the selection of the optimal model. For example, for models $A$, $B$, and $C$, we aim to determine whether their capability values are [0.9, 0.41, 0.4] or [0.9, 0.89, 0.4], rather than just concluding that $A > B > C$. This distinction is crucial in practical applications because model deployment decisions often involve balancing performance and cost. If $A$’s API price is significantly higher than $B$’s and we know with precision that the performance gap between $A$ and $B$ is minimal (e.g., 0.9 vs. 0.89), we might prefer model $B$. Without precise capability values, such decisions become difficult when only ranking information is available.
>
> Therefore, when our optimization goal is to determine the exact capability values of models, uniform sampling becomes intuitive. First, since tasks vary in difficulty, the same model may perform differently across tasks. As mentioned on line 213 in the original manuscript, the transitivity of model performance may not hold in some cases ($A> B, B>C, C>A$). Uniform sampling across candidates helps mitigate these biases and improves accuracy. Second, since we need the capability values for all models, uniformly sampling across them ensures balanced data collection, reducing the uncertainty in the estimated capability values for any particular model.
>
> **References**
>
> [1] Jin Peng Zhou, Christian K. Belardi, Ruihan Wu, Travis Zhang, Carla P. Gomes, Wen Sun, and
> Kilian Q. Weinberger. On speeding up language model evaluation.

---

> > ### Comment · Reviewer_8sRK · 2024-11-22
> >
> > Thanks for the response, but my question is not fully addressed: The authors argue that to avoid bias, the budget should be allocated uniformly, if so how could this method be more sample efficient than random?

---

> ### Author Response · Authors · 2024-11-22
> **Further Clarification**
>
> We apologies for our earlier focus on explaining why the budget should be allocated uniformly. Now, let us address why UniCBE outperforms Random. (We have also provided a PDF version via an anonymous link for a clearer and more complete presentation of our response: https://anonymous.4open.science/r/Respones-to-Reviewer-8sRK-8A44/FURTHER%20DISCUSSION%20WITH%20REVIEWER%208SRK.pdf)
>
> At the expectation level, random sampling ensures a relatively uniform distribution across different tuple combinations, thereby guaranteeing the accuracy of evaluation results (lines 108-110). However:
>
> - **Argument 1**: On the one hand, randomness prevents the optimization objective from being fully achieved, meaning uniformity cannot be attained optimally.
> - **Argument 2**: On the other hand, UniCBE’s optimization objectives go beyond improving accuracy to include enhanced scalability and convergence, which random sampling cannot achieve.
>
> **Experimental Validation:**
>
> As shown in Table below, the $\beta_{\text{acc}}$ achieved by RANDOM is lower than that of UniCBE, which validates that randomness results in suboptimal uniformity (Argument 1). Additionally, the $\beta_{\text{sca}}$ and $\beta_{\text{con}}$ achieved by RANDOM are also lower than those of UniCBE, confirming that UniCBE explicitly optimizes for scalability and convergence objectives, unlike Random (Argument 2).
> | Methods      |    Random | UniCBE  |
> | :-------- | --------:| :--: |
> | $\beta_{acc}$  | .5803 |  .7364   |
> | $\beta_{con}$    |  .9081 |  .9228  |
> | $\beta_{sca}$ |  .9972 | .9997  |
>
> Furthermore, we experimentally validated this from another perspective involving temperature. We reformulate sampling matrix $P^l$ to:
>
> $$ f^{ts}_T(i,j,k) $$
>
> $$ =\frac{(P^l_{i,j,k})^{-T}}{\sum (P^{l})^{-T}} $$
>
> By adjusting the temperature $T$, we control the extent of uniformity constraints. As $T$ increases, the uniformity constraints are progressively relaxed. When $T=0$, it corresponds to UniCBE (greedy sampling), which enforces the strictest uniformity constraints. When $T = +\infty$, it corresponds to random sampling, where no uniformity constraints are applied. As illustrated in Figure 10, as $T$ increases from $0$ to $+\infty$, the evaluation results progressively deteriorate. This demonstrates that adopting greedy sampling to enforce the strictest uniformity constraints yields the best evaluation performance, further validating Argument 1.
>
> **We sincerely apologize once again for the oversight in our previous response! If you have any questions, we would be more than happy to engage in further communication with you. At the same time, we would like to express our heartfelt gratitude for your support and suggestions regarding our work.**

---

> > ### Comment · Reviewer_8sRK · 2024-11-22
> >
> > Thanks for your explanation! I will keep my score.

---

### Official Review · Reviewer_8LCX · 2024-11-02

**Soundness:** 4
**Presentation:** 4
**Contribution:** 4
**Rating:** 8
**Confidence:** 4

**Summary:**

The paper presents UNICBE, a uniformity-driven comparing-based evaluation (CBE) framework designed to optimize model evaluation across three primary objectives: accuracy, convergence, and scalability. Existing CBE method such as random allocation, method used in ARENA and ALPACAEVAL fails to maximize these aspects simultaneously.
UNICBE addresses this by simple yet effective approach to promote uniformity across sample and models. The proposed method incorporates three decoupled sampling probability matrices, each of them derived to ensure uniformity in accuracy, convergence, scalability to newly added models. Comprehensive experiments and ablation studies across multiple datasets highlight UNICBE's cost-efficient performance in model evaluation.

**Strengths:**

UNICBE introduces a novel approach to CBE by balancing three critical objectives, accuracy, convergence, and scalability, representing a straightforward yet impactful optimization techniques to advance model evaluation.
While the framework’s calculations and derivations to promote uniformity are relatively simple, its contributions are substantial given the current reliance on labor-intensive evaluation processes.
The quality of the work is supported by extensive empirical analysis across diverse models, benchmarks, and settings, with extensive testing to support the assumptions in Section 3 and performance validation across multiple configurations in Section 5.4.
Clarity is evident in the structured presentation of key concepts and experimental results, with details in experiment or notations clearly addressed. This paper's significance lies in its contribution to large language model evaluation, providing a scalable and efficient methodology that aligns closely with human preference signals, a crucial advancement for iterative model assessments.

**Weaknesses:**

The novelty of balancing accuracy, convergence, and scalability needs further justification, as similar uniform sampling strategies have been discussed in prior works that highlight the uniformity, such as Vabalas et al. (2019) for sampling biases, which could diminish its uniqueness.

Although the experiment of MT-Bench is based on human evaluator, larger portion of the evaluation is relied on AlpacaEval, as larger number of models and samples are used for the evaluation with AlpacaEval. The reliance on GPT-4 and GPT-3.5-turbo as evaluators, while useful, could benefit from validation against human judgments or additional LLMs, such as Claude, to establish greater reliability and generalizability across evaluator types.

Minor details, but the readability of all figures could be enhanced by widening the lines in each plot, which would improve clarity and interpretation for readers.

**Questions:**

As the UniCBE is based on three matrix, $P^{acc-l}, P^{con-l}, P^{sca-l}$, each targeting different goal of accuracy, convergence, scaliability, can user steer between those by adding hyperparameter for each matrix? Would it be also possible to quantify it through experiment?

While scalability is addressed by sequentially adding models, the paper could enhance this section by incorporating real-world scenarios, where models enter and exit dynamically, further proving UNICBE’s robustness in evolving benchmarks.

The given choice of greedy sampling over probabilistic sampling and Bradley-Terry model over Elo rating system appears significant to the framework’s success. Could the authors conduct a small experiment to demonstrate that UniCBE maintains its effectiveness across different sampling and aggregation settings?

---

> ### Author Response · Authors · 2024-11-19
> **Response to Reviewer 8LCX**
>
> Dear Reviewer 8LCX, we sincerely thank you for your valuable feedback on our submission. Below is our responses to the concerns you raised. We have incorporated these contents into the updated version of our paper, which we believe will help enhance the quality of our submission.
>
> **Since this page cannot display images, we have provided a PDF version via an anonymous link for a clearer and more complete presentation of our response: https://anonymous.4open.science/r/Respones-to-Reviewer-8LCX-809F/response%20to%20reviewer%208LCX.pdf. This can also be find in the Supplementary Material. We recommend viewing the PDF in the provided link for easier reading.**
>
> > **The novelty of balancing accuracy, convergence, and scalability needs further justification, as similar uniform sampling strategies have been discussed in prior works that highlight the uniformity, such as Vabalas et al. (2019) for sampling biases, which could diminish its uniqueness.**
>
> We agree with your suggestion to discuss related work concerning sampling uniformity in connection with our work. Here is our discussion:
>
> Previous studies have discussed the risks of introducing sampling bias in incomplete sampling scenarios. Specifically, [1] demonstrated through simulation experiments that K-fold cross-validation (K-fold CV) can produce significant performance estimation bias when dealing with small sample sizes. This bias persists even when the sample size reaches 1000. In contrast, methods like nested cross-validation (Nested CV) and train/test split have been shown to provide robust and unbiased performance estimates regardless of sample size. [2] introduced a weighting scheme, as described in [3], to mitigate sampling bias in active testing scenarios. [4] proposed leveraging information obtained from source models to select representative samples from the test set, thereby reducing sampling bias. Additionally, [5] employed Item Response Theory [6] to correct sample bias in addressing this issue.
>
> These studies inspired us to investigate the bias problem in the CBE scenario. Unlike the aforementioned studies, we found that in CBE scenario, not only does sample bias exist, but model bias also plays a role, and the two are coupled. This coupling poses greater challenges for analyzing and mitigating these biases. To address this, based on the analyses outlined in Section 3, we propose the UniCBE method, which effectively alleviates biases in this scenario.
>
> > **Although the experiment of MT-Bench is based on human evaluator, larger portion of the evaluation is relied on AlpacaEval, as larger number of models and samples are used for the evaluation with AlpacaEval. The reliance on GPT-4 and GPT-3.5-turbo as evaluators, while useful, could benefit from validation against human judgments or additional LLMs, such as Claude, to establish greater reliability and generalizability across evaluator types.**
>
> Yes, in our experiments, we test the performance of UniCBE with humans, GPT-4o, and GPT-3.5-turbo as judges. We recognize that involving a broader range of evaluators helps enhance the robustness and generalizability of our experimental conclusions. Therefore, below we additionally test the compared methods using Qwen-Plus[7] as the judge. As shown in Figure 14, when Qwen-Plus is used as the judge, the experimental results are similar to those obtained with other types of judges. UniCBE achieves significant preference budget savings, exceeding 20%. This experimental result further validates the generalizability of UniCBE to different sources of preference signals.
>
> > **Minor details, but the readability of all figures could be enhanced by widening the lines in each plot, which would improve clarity and interpretation for readers.**
>
> Thank you for your valuable suggestions. We have increased the width of the lines in the figure in the revised version to enhance readability. You can refer to the latest version.
>
> **References**
>
> [1] Sebastian Farquhar, Yarin Gal, and Tom Rainforth. On statistical bias in active learning: How and
> when to fix it.
>
> [2] Jannik Kossen, Sebastian Farquhar, Yarin Gal, et al. Active testing: Sample-efficient
> model evaluation.
>
> [3] Frederic M Lord and Melvin R Novick. Statistical theories of mental test scores.
>
> [4] Felipe Maia Polo, Lucas Weber, Leshem Choshen, et al. tinybenchmarks: evaluating llms with fewer examples.
>
> [5] Andrius Vabalas, Emma Gowen, Ellen Poliakoff, et al. Machine learning algorithm validation with a limited sample size.
>
> [6] Rajan Vivek, Kawin Ethayarajh, Diyi Yang, et al. Anchor points: Benchmarking models with much fewer examples.
>
> [7] An Yang, Baosong Yang, Binyuan Hui, et al. Qwen2 technical report.

---

> ### Author Response · Authors · 2024-11-19
> **Response to Reviewer 8LCX (part 2)**
>
> > **As the UniCBE is based on three matrix, each targeting different goal of accuracy, convergence, scalability, can user steer between those by adding hyperparameter for each matrix? Would it be also possible to quantify it through experiment?**
>
> We agree with your suggestion to add hyperparameters for each matrix to achieve controllability for different optimization objectives.
> In the original manuscript, we integrate sampling matrices targeting different optimization objectives with equal weights:
> $$P^{l} = \frac{P^{acc\text{-}l} \circ P^{con\text{-}l} \circ P^{sca\text{-}l}}{\sum (P^{acc\text{-}l} \circ P^{con\text{-}l} \circ P^{sca\text{-}l})}$$
> In practice, when faced with varying requirements, it is straightforward to prioritize a specific objective by adjusting the weights \(\theta_{acc}\), \(\theta_{con}\), and \(\theta_{sca}\) for these matrices, as shown below:
> $$P^{l} = \frac{(P^{acc\text{-}l})^{\theta_{acc}} \circ (P^{con\text{-}l})^{\theta_{con}}  \circ (P^{sca\text{-}l})^{\theta_{sca}} }{\sum ((P^{acc\text{-}l})^{\theta_{acc}} \circ (P^{con\text{-}l})^{\theta_{con}}  \circ (P^{sca\text{-}l})^{\theta_{sca}})}$$
> As demonstrated in Table below, we set different settings and calculate the degree of achievement level for each optimization objective $\beta$ following the calculation procedure described in Appendix-E. Compared to equal-weight integration, users can easily increase the corresponding $\beta$ (e.g., $\beta_{acc}$) by assigning a larger weight to a specific optimization objective ($\theta_{acc}$), thereby better meeting their practical needs (accuracy).
> We also observe that enhancing a specific optimization objective often comes with a slight decrease in the achievement of other objectives. In Figure 11, we illustrate an example of improving accuracy, where $\theta_{acc}$ is increased from 1 to 2. We find that the increased focus on accuracy objective slightly slows down the convergence speed. As a result, when $T$ is relatively small, the performance of $\theta_{acc} = 2$ lags behind that of $\theta_{acc} = 1$. However, in the later stages, after convergence, the enhanced accuracy objective enables $\theta_{acc} = 2$ to outperform $\theta_{acc} = 1$, resulting in greater savings in the preference budget.
>
> | Settings      |   $\theta_{acc} = 2, \theta_{con} = 1,\theta_{sca} = 1$ | $\theta_{acc} = 1, \theta_{con} = 2,\theta_{sca} = 1$  |$\theta_{acc} = 1, \theta_{con} = 1,\theta_{sca} = 2$|$\theta_{acc} = 1, \theta_{con} = 1,\theta_{sca} = 1$|
> |:-:|:-:|:-:|:-:|:-:|
> |$\beta_{acc}$|**.7380**|.7355|.7351|.7364|
> |$\beta_{con}$|.9221|**.9235**|.9217|.9228|
> |$\beta_{sca}$|.9996|.9997|**.9998**|.9997|
>
> > **While scalability is addressed by sequentially adding models, the paper could enhance this section by incorporating real-world scenarios, where models enter and exit dynamically, further proving UniCBE’s robustness in evolving benchmarks.**
>
> We agree with your suggestion that testing UniCBE in a highly dynamic, real-time evaluation setting can help us more comprehensively assess its performance. To this end, we conduct the following experiments:
> Starting with a sample size of $N=600$ and model number of $M=12$, we execute a random operation at each time step. The operations included: adding one model to be evaluated with a probability of 0.01, removing one model with a probability of 0.01, adding one potential sample with a probability of 0.01, randomly deleting one sample with a probability of 0.01, and taking no action with a probability of 0.96. Based on the experimental results shown in Figure 9, we have the following observations:
> - The convergence speed of all baseline methods significantly slowed down. None of the baseline methods achieve a Spearman correlation coefficient of 0.96 or a Pearson correlation coefficient of 0.97 by $T=2000$, highlighting the difficulty of model evaluation in this setting. In contrast, UniCBE achieve rapid convergence, reaching a Spearman coefficient of approximately 0.97 and a Pearson coefficient exceeding 0.98 by $T=2000$.
> - Over the long term, as $T$ increases, UniCBE consistently demonstrates over 10% savings in preference budget across all metrics, even under this challenging setting, showcasing its strong practicality.
> - An interesting observation is that AlpacaEval exhibits better convergence in the early stages compared to Random and Arena, supporting our previous conclusions in Table 1. However, as $T$ increases, AlpacaEval's lack of accuracy optimization objective leads to its performance being surpassed by Random and Arena.

---

> ### Author Response · Authors · 2024-11-19
> **Response to Reviewer 8LCX (part 3)**
>
> > **The given choice of greedy sampling over probabilistic sampling and Bradley-Terry model over Elo rating system appears significant to the framework’s success. Could the authors conduct a small experiment to demonstrate that UniCBE maintains its effectiveness across different sampling and aggregation settings?**
>
> We appreciate your valuable suggestion. In fact, as shown in Figure 5, we explore the combination of UniCBE with probability sampling strategy, Elo rating and average win rate aggregation strategies, comparing these with the default configuration. Our findings are as follows:
>
> - UniCBE consistently outperforms the baselines when combined with various settings.
> - Under the default configuration (greedy sampling and BT model), UniCBE achieves optimal performance. We believe this is because greedy sampling maximizes sampling uniformity, and the BT model better alleviates sampling bias in cases of misaligned samples.

---

> ### Author Response · Authors · 2024-11-25
> **Kindly Reminder**
>
> Dear Reviewer 8LCX,
>
> We sincerely appreciate your recognition of our work, UniCBE, as well as the time and effort you have dedicated to improving its quality. We have carefully addressed the questions and suggestions you raised, and have incorporated these valuable insights into the revised version, which we believe will further enhance the completeness of the manuscript and the generalizability of its conclusions. Noting that the discussion phase is approaching its end (11/26, AOE), we would be more than happy to engage in further discussion should you have any additional suggestions or questions.
>
> Best regards.

---

> > ### Comment · Reviewer_8LCX · 2024-11-26
> >
> > Thank you for your detailed response. The additional experiment with varied hyperparameters enhances the reliability of the paper's main contribution. I have updated my score accordingly.

---

### Official Review · Reviewer_PBe8 · 2024-11-03

**Soundness:** 3
**Presentation:** 3
**Contribution:** 3
**Rating:** 6
**Confidence:** 3

**Summary:**

This paper studies to explore the use of preference signals with comparing-based evaluation (CBE). It proposes a unified uniformity-driven framework that can achieve CBE with better accuracy. Experiments show the proposed method saves over 17% of evaluation budgets compared to the random sampling baseline.

**Strengths:**

1. The study of using preference signals with lower evaluation budgets is important research direction with huge potentials for benefiting future research and development

2. This paper proposes a framework to achieve CBE with better accuracy, convergence and scalability.

**Weaknesses:**

1. The paper can be improved with more performance comparison with existing work and state-of-the-art performance.

2. Expect more statistical and experimental conclusions with the proposed CBE method for the scenario of large-scale preference learning.

**Questions:**

None.

---

> ### Author Response · Authors · 2024-11-19
> **Response to Reviewer PBe8**
>
> Dear Reviewer PBe8, we sincerely thank you for your valuable feedback on our submission. Below is our responses to the concerns you raised. We have incorporated these contents into the updated version of our paper, which we believe will help enhance the quality of our submission.
>
> **Since this page cannot display images, we have provided a PDF version via an anonymous link for a clearer and more complete presentation of our response: https://anonymous.4open.science/r/Respones-to-Reviewer-PBe8-1DB1/response%20to%20reviewer%20PBe8.pdf. This can also be find in the Supplementary Material. We recommend viewing the PDF in the provided link for easier reading.**
>
> > **The paper can be improved with more performance comparison with existing work and state-of-the-art performance.**
>
> We agree with your suggestion that comparing UniCBE with state-of-the-art methods can help fully validate its effectiveness. In our experiments, we select the most widely used state-of-the-art methods, Arena and AlpacaEval, for comparison to evaluate the performance of UniCBE. In fact, although comparing-based evaluation holds significant research value in the era of large language models, the existing research in this area remains limited. Arena and AlpacaEval represent the most relevant and practical works we could identify. We encourage future researchers to explore this field further and investigate more efficient ways to utilize valuable preference signals.
>
> > **Expect more statistical and experimental conclusions with the proposed CBE method for the scenario of large-scale preference learning.**
>
> We agree with your suggestion and therefore attempt to conduct experiments in a large-scale, highly dynamic CBE scenario that is closer to real-world conditions. Specifically:
> Starting with a sample size of $N=600$ and model number of $M=12$, we execute a random operation at each time step. The operations included: adding one model to be evaluated with a probability of 0.01, removing one model with a probability of 0.01, adding one potential sample with a probability of 0.01, randomly deleting one sample with a probability of 0.01, and taking no action with a probability of 0.96. Based on the experimental results shown in Figure 9, we have the following observations:
> - The convergence speed of all baseline methods significantly slowed down. None of the baseline methods achieve a Spearman correlation coefficient of 0.96 or a Pearson correlation coefficient of 0.97 by $T=2000$, highlighting the difficulty of model evaluation in this setting. In contrast, UniCBE achieve rapid convergence, reaching a Spearman coefficient of approximately 0.97 and a Pearson coefficient exceeding 0.98 by $T=2000$.
> - Over the long term, as $T$ increases, UniCBE consistently demonstrates over 10% savings in preference budget across all metrics, even under this challenging setting, showcasing its strong practicality.
> - An interesting observation is that AlpacaEval exhibits better convergence in the early stages compared to Random and Arena, supporting our previous conclusions in Table 1. However, as $T$ increases, AlpacaEval's lack of accuracy optimization objective leads to its performance being surpassed by Random and Arena.

---

> > ### Comment · Reviewer_PBe8 · 2024-11-25
> > **After rebuttal**
> >
> > I sincerely thank the authors for answering my questions, and I appreciate the tremendous and excellent work for preparing the paper and addressing the questions from all reviewers. I am still curious about the method's practical performance on large-scale learning. The scale in the experiment above is not enough to address my concerns. Thus, I will keep my original rating.
> >
> > Thanks for choosing to publish this excellent work!

---

> ### Author Response · Authors · 2024-11-25
> **Kindly Reminder**
>
> Dear Reviewer PBe8,
>
> We sincerely appreciate your recognition of our work, UniCBE, as well as the time and effort you have dedicated to improving its quality. We have carefully addressed the questions and suggestions you raised, and have incorporated these valuable insights into the revised version, which we believe will further enhance the completeness of the manuscript and the generalizability of its conclusions. Noting that the discussion phase is approaching its end (11/26, AOE), we would be more than happy to engage in further discussion should you have any additional suggestions or questions.
>
> Best regards.

---

### Official Review · Reviewer_k9DA · 2024-11-03

**Soundness:** 3
**Presentation:** 2
**Contribution:** 3
**Rating:** 8
**Confidence:** 4

**Summary:**

The paper presents UniCBE, a uniformity-driven comparing-based evaluation (CBE) framework designed to improve the efficiency and accuracy of evaluating large language models (LLMs) based on human preferences. UniCBE optimizes three main objectives simultaneously: reducing sampling bias, enhancing convergence by managing uncertainty, and ensuring scalability when new models are introduced. Compared to baselines like random, Arena, and AlpacaEval, UniCBE demonstrates the lowest error and highest correlation between the ground truth evaluation results.

**Strengths:**

- Comparing-based evaluation is an important problem in LLM evaluation. The proposed method solved a non-trivial problem.
- This paper improves the comparing-based evaluation from three perspectives (accuracy, convergence, and stability). It provides a solid theory foundation, and the experiment results also demonstrate the efficiency of its proposed method.
- Their experiments cover three dimensions well in their proposed method. They also provide ablation studies on each variant. In general, most of the variants play their roles well.

**Weaknesses:**

- The motivation is not enough clear. Why the previous method cannot perform well in both accuracy, convergence, and scalability?
- The runtime of the previous CBE method ($O(NM^2)$) is one of the major limitations, and the author starts from this limitation as one of the motivations for the proposed method. However, they lack the runtime analysis for the UniCBE but only an approximate number for saving time when compared to the previous method.
- While UniCBE shows promising results for scenarios with periodically introduced new models, it may be less efficient in highly dynamic, real-time evaluation settings where new models or samples are constantly introduced at high frequencies.

**Questions:**

See Weakness.

---

> ### Author Response · Authors · 2024-11-19
> **Response to Reviewer k9DA**
>
> Dear Reviewer k9DA, we sincerely thank you for your valuable feedback on our submission. Below is our responses to the concerns you raised. We have incorporated these contents into the updated version of our paper, which we believe will help enhance the quality of our submission.
>
> **Since this page cannot display images, we have provided a PDF version via an anonymous link for a clearer and more complete presentation of our response: https://anonymous.4open.science/r/Respones-to-Reviewer-k9DA-4842/response%20to%20reviewer%20k9DA.pdf. This can also be find in the Supplementary Material. We recommend viewing the PDF in the provided link for easier reading.**
>
> > **The motivation is not enough clear. Why the previous method cannot perform well in both accuracy, convergence, and scalability?**
>
> Previous methods often focus on a single optimization objective (lines 64–65, Table 1) while neglecting the optimization of other attributes. In the related work section (lines 107–118), we specifically explain that the Random, Arena, and AlpacaEval methods each focus solely on optimizing accuracy, convergence, or scalability, respectively. Furthermore, we demonstrate in Table 2 through experiments that these three methods result in unbalanced sampling across multiple dimensions due to their failure to simultaneously consider all three optimization objectives. This ultimately leads to their inability to perform well across all three attributes simultaneously.
>
> > **The runtime of the previous CBE method ($O(NMM)$) is one of the major limitations, and the author starts from this limitation as one of the motivations for the proposed method. However, they lack the runtime analysis for the UniCBE but only an approximate number for saving time when compared to the previous method.**
>
> We fully agree with your suggestion to include an analysis of UniCBE's runtime. To address this, we provide the following statistics: when a win rate error $\Delta$ of less than 0.02 is required, UniCBE needs a preference budget of $T = 2800$ (compared to $T = 3200$ for Random). Similarly, when $\Delta$ is less than 0.01, UniCBE requires $T = 9400$ (compared to $T = 11300$ for Random). Additionally, the computational complexity of $O(NMM)$ is $N \cdot M \cdot (M - 1) / 2 = 84525$. Therefore, under specific evaluation accuracy requirements, UniCBE significantly reduces the preference budget needed.
>
> > **While UniCBE shows promising results for scenarios with periodically introduced new models, it may be less efficient in highly dynamic, real-time evaluation settings where new models or samples are constantly introduced at high frequencies.**
>
> We believe that testing UniCBE in a highly dynamic, real-time evaluation setting can help us more comprehensively assess its performance. To this end, we conduct the following experiments:
> Starting with a sample size of $N=600$ and model number of $M=12$, we execute a random operation at each time step. The operations included: adding one model to be evaluated with a probability of 0.01, removing one model with a probability of 0.01, adding one potential sample with a probability of 0.01, randomly deleting one sample with a probability of 0.01, and taking no action with a probability of 0.96. Based on the experimental results shown in Figure 9, we have the following observations:
> - The convergence speed of all baseline methods significantly slowed down. None of the baseline methods achieve a Spearman correlation coefficient of 0.96 or a Pearson correlation coefficient of 0.97 by $T=2000$, highlighting the difficulty of model evaluation in this setting. In contrast, UniCBE achieve rapid convergence, reaching a Spearman coefficient of approximately 0.97 and a Pearson coefficient exceeding 0.98 by $T=2000$.
> - Over the long term, as $T$ increases, UniCBE consistently demonstrates over 10% savings in preference budget across all metrics, even under this challenging setting, showcasing its strong practicality.
> - An interesting observation is that AlpacaEval exhibits better convergence in the early stages compared to Random and Arena, supporting our previous conclusions in Table 1. However, as $T$ increases, AlpacaEval's lack of accuracy optimization objective leads to its performance being surpassed by Random and Arena.

---

> > ### Comment · Reviewer_k9DA · 2024-11-20
> >
> > I'm satisfied with the authors' reply regarding the runtime and real-time analysis, so I adjusted my score accordingly.

---

### Official Review · Reviewer_D5Wd · 2024-11-04

**Soundness:** 3
**Presentation:** 3
**Contribution:** 3
**Rating:** 6
**Confidence:** 4

**Summary:**

This paper introduces UNICBE, a uniformity-driven framework for comparing-based evaluation (CBE) that simultaneously optimizes multiple objectives in language model assessment. Current CBE methods suffer from limitations related to bias, slow convergence, and lack of scalability.

**Strengths:**

Efficient evaluation methods are essential for increasingly large-scale LLMs, and this work addresses relevant limitations in conventional CBE.

The integration of accuracy, convergence, and scalability in a uniformity-driven approach is a valuable contribution, setting UNICBE apart from single-objective methods.

**Weaknesses:**

The framework assumes preference signals (particularly from automated judges like GPT-4) are consistent with human judgment, a potentially risky simplification given known limitations in automated preference evaluations.

The formulation of multi-dimensional sampling matrices and their interaction in optimizing accuracy, convergence, and scalability may be overly complex for practical implementations and difficult to interpret for further tuning or adjustment.

**Questions:**

1. How does UNICBE perform when preference signals are less reliable, as is often the case with models lower than GPT-4 or inconsistent human annotations?

2. Could the authors elaborate on how UNICBE would handle scenarios with dynamic preference priorities, where, for example, accuracy is weighted more heavily than convergence?


3. To what extent could the uniformity constraints in the sampling matrices be relaxed while maintaining cost-effectiveness?

---

> ### Author Response · Authors · 2024-11-19
> **Response to Reviewer D5Wd**
>
> Dear Reviewer D5Wd, we sincerely thank you for your valuable feedback on our submission.
> Below is our responses to the concerns you raised.
> We have incorporated these contents into the updated version of our paper, which we believe will help enhance the quality of our submission.
>
> **Since this page cannot display images, we have provided a PDF version via an anonymous link for a clearer and more complete presentation of our response: https://anonymous.4open.science/r/Respones-to-Reviewer-D5Wd-2AC1/response%20to%20reviewer%20D5Wd.pdf. This can also be find in the Supplementary Material.
> We recommend viewing the PDF in the provided link for easier reading.**
>
> > **The framework assumes preference signals (particularly from automated judges like GPT-4) are consistent with human judgment, a potentially risky simplification given known limitations in automated preference evaluations.**
>
> We understand your concern about the existence of bias between preference signals from models and humans.
> In fact, we don't think the preference signals from LLMs are consistent with human judgment .
> We experiment with both LLMs and human as judges for the following two reasons:
> - First, high-quality benchmarks that align human supervisory signals at both the sample and model levels in CBE scenario are relatively limited, so we focus on evaluating the effectiveness of UniCBE under human preference signals using MT-Bench.
> - Second, as an increasing number of studies (e.g., AlpacaEval) begin to adopt LLMs as judges, it is also important to validate the effectiveness of UniCBE under model preference signals.
>
> Experimental results show that UniCBE demonstrates superior performance when using GPT-4o, GPT-3.5-turbo, and humans as judges, confirming the robustness of UniCBE to different sources of preference signals.
>
> > **The formulation of multi-dimensional sampling matrices and their interaction in optimizing accuracy, convergence, and scalability may be overly complex for practical implementations and difficult to interpret for further tuning or adjustment. Could the authors elaborate on how UniCBE would handle scenarios with dynamic preference priorities, where, for example, accuracy is weighted more heavily than convergence?**
>
> In the original manuscript, we integrate sampling matrices targeting different optimization objectives with equal weights:
> $$P^{l} = \frac{P^{acc\text{-}l} \circ P^{con\text{-}l} \circ P^{sca\text{-}l}}{\sum (P^{acc\text{-}l} \circ P^{con\text{-}l} \circ P^{sca\text{-}l})}$$
> In practice, when faced with varying requirements, it is straightforward to prioritize a specific objective by adjusting the weights \(\theta_{acc}\), \(\theta_{con}\), and \(\theta_{sca}\) for these matrices, as shown below:
> $$P^{l} = \frac{(P^{acc\text{-}l})^{\theta_{acc}} \circ (P^{con\text{-}l})^{\theta_{con}}  \circ (P^{sca\text{-}l})^{\theta_{sca}} }{\sum ((P^{acc\text{-}l})^{\theta_{acc}} \circ (P^{con\text{-}l})^{\theta_{con}}  \circ (P^{sca\text{-}l})^{\theta_{sca}})}$$
> As demonstrated in Table below, we set different settings and calculate the degree of achievement level for each optimization objective $\beta$ following the calculation procedure described in Appendix-E. Compared to equal-weight integration, users can easily increase the corresponding $\beta$ (e.g., $\beta_{acc}$) by assigning a larger weight to a specific optimization objective ($\theta_{acc}$), thereby better meeting their practical needs (accuracy).
> We also observe that enhancing a specific optimization objective often comes with a slight decrease in the achievement of other objectives. In Figure 11, we illustrate an example of improving accuracy, where $\theta_{acc}$ is increased from 1 to 2. We find that the increased focus on accuracy objective slightly slows down the convergence speed. As a result, when $T$ is relatively small, the performance of $\theta_{acc} = 2$ lags behind that of $\theta_{acc} = 1$. However, in the later stages, after convergence, the enhanced accuracy objective enables $\theta_{acc} = 2$ to outperform $\theta_{acc} = 1$, resulting in greater savings in the preference budget.
> | Settings      |   $\theta_{acc} = 2, \theta_{con} = 1,\theta_{sca} = 1$ | $\theta_{acc} = 1, \theta_{con} = 2,\theta_{sca} = 1$  |$\theta_{acc} = 1, \theta_{con} = 1,\theta_{sca} = 2$|$\theta_{acc} = 1, \theta_{con} = 1,\theta_{sca} = 1$|
> |:-:|:-:|:-:|:-:|:-:|
> |$\beta_{acc}$|**.7380**|.7355|.7351|.7364|
> |$\beta_{con}$|.9221|**.9235**|.9217|.9228|
> |$\beta_{sca}$|.9996|.9997|**.9998**|.9997|

---

> ### Author Response · Authors · 2024-11-19
> **Response to Reviewer D5Wd (part 2)**
>
> > **How does UniCBE perform when preference signals are less reliable, as is often the case with models lower than GPT-4 or inconsistent human annotations?**
>
> As shown in Figure 6, in addition to GPT-4o and humans, we also conduct experiments using GPT-3.5-turbo as the judge, whose preference signals are less reliable. The experiments (lines 463–466) demonstrate a noticeable decline in the performance of all methods (particularly, the Arena method performs almost on par with random sampling), which is likely due to the increased noise in the preferences provided by GPT-3.5-turbo, leading to slower convergence. In comparison, UniCBE still achieves over a 15% preference budget savings relative to random sampling.
>
> > **To what extent could the uniformity constraints in the sampling matrices be relaxed while maintaining cost-effectiveness?**
>
> Based on our analyses in Section 3, the degree to which uniformity is achieved is positively correlated with performance in terms of accuracy, convergence, and scalability. To explore the empirical relationship between the degree of uniformity constraints and the final outcomes, we draw inspiration from the concept of temperature-based control in random sampling. By adjusting the temperature $T$ in the following formula for sampling $f^{ts}_T$, we regulate the extent of uniformity constraints:
>
> $$f^{ts}_T(i,j,k)$$
>
> $$=\frac{(P^l_{i,j,k})^{-T}}{\sum (P^{l})^{-T}}$$
>
> As $T$ increases, the uniformity constraints become more relaxed. When $T=0$, it corresponds to greedy sampling $f^{ts}_g$, which imposes the strictest uniformity constraints. When $T=1$, it corresponds to probabilistic sampling $f^{ts}_p$, which imposes general uniformity constraints. When $T=+\infty$, it corresponds to random sampling, where no uniformity constraints are applied.
> Our experimental results are shown in Figure 10.
> As $T$ increases from 0 to $+\infty$, the evaluation results progressively deteriorate. This indicates that adopting greedy sampling to impose the strictest uniformity constraints yields the optimal evaluation performance. This observation also validates the correctness of our conclusions in Section 3.

---

> > ### Comment · Reviewer_D5Wd · 2024-11-21
> >
> > Some of my concerns have been addressed, so I have increased my score.

---

### Author Response · Authors · 2024-12-02
**Sincere Gratitude to All the Reviewers**

**We sincerely thank all the reviewers for their time and effort in reviewing our manuscript. All the reviewers provided valuable suggestions and actively engaged in timely and constructive discussions, setting an excellent example for our community.** Based on their valuable suggestions, we have incorporated the following updates into the revised version:

- Validated the performance of UniCBE under more dynamic and realistic scenarios, while extending the preference budget from 8000 to 10000. This provides stronger support for the application of UniCBE in large-scale real-world scenarios.
- Investigated the performance of UniCBE under varying levels of uniformity constraints. This serves as a more fine-grained ablation study to verify the validity of the uniformity optimization objective.
- Explored the effects of adjusting the weights of different optimization objectives, demonstrating the flexibility of UniCBE and offering insights for more customized use cases.
- Included experimental results using Qwen-Plus as the judge, further validating the generalization capability of UniCBE.
- Expanded the discussion on related works concerning sampling bias, which supports and clarifies the motivation of our study.
- Elaborated on the intuitive motivation behind UniCBE, making it easier for future readers to understand our work.

We believe these additions have significantly improved the quality of our work. We also look forward to seeing more researchers explore comparison-based evaluation in the future, achieving ongoing progress in the efficiency and accuracy of evaluation signals.

---

### Meta-Review · Area_Chair_puX9 · 2024-12-23

**Metareview:**

This paper introduces a comparing-based evaluation (CBE) framework which considers at the same time multiple objectives, such that it can leverage scarce signals. In that way, it can become more efficient while considering the standard factors of LLM CBE: accuracy, convergence and scalability.

The paper is well motivated: it addresses a timely and important problem in a way that intuitively makes sense. This research also opens up the way for further work for LLM CBE under this new viewpoint.

This paper has been reviewed by five experts and it has benefited from fruitful discussion between authors and reviewers. A few initial concerns were clarified during the rebuttal, as it was acknowledged by the reviewers. The only substantial remaining concern is about the scale of the experiments being relatively small; however, overall the experiments have been quite satisfactory.

Therefore, overall this paper contains solid research and it will likely be of interest to the ICLR community.

**Additional Comments On Reviewer Discussion:**

Several clarifications were offered by the authors during the rebuttal period, and the reviewers have acknowledged that these largely address their concerns. The authors also offered new experimental results which have been welcomed by the reviewers.

---

### Decision · Program_Chairs · 2025-01-22

Accept (Spotlight)